# JMJD3 and UTX determine fidelity and lineage specification of human neural progenitor cells

Yongli Shan[1,2,3,4,5,8], Yanqi Zhang[1,3,4,5,8], Yuan Zhao[1,3,4,5,8], Tianyu Wang [1,4,5], Jingyuan Zhang[1,3,4,5], Jiao Yao[1,4,5], Ning Ma[1], Zechuan Liang[1,4,5], Wenhao Huang[1,4,5], Ke Huang[1,4,5], Tian Zhang[1,3,4,5], Zhenghui Su[1,4,5], Qianyu Chen[1,4,5], Yanling Zhu[1,3,4,5], Chuman Wu[1,4,5], Tiancheng Zhou[1,4,5], Wei Sun[1,4,5], Yanxing Wei[2], Cong Zhang[1,3,4,5], Chenxu Li[1,4], Shuquan Su[1,4], Baojian Liao[1,4,5], Mei Zhong[2], Xiaofen Zhong[1,4,5], Jinfu Nie[1,4,5], Duanqing Pei [1,4,5] & Guangjin Pan[1,4,5,6,7]*

Neurogenesis, a highly orchestrated process, entails the transition from a pluripotent to neural state and involves neural progenitor cells (NPCs) and neuronal/glial subtypes. However, the precise epigenetic mechanisms underlying fate decision remain poorly understood. Here, we delete KDM6s (JMJD3 and/or UTX), the H3K27me3 demethylases, in human embryonic stem cells (hESCs) and show that their deletion does not impede NPC generation from hESCs. However, KDM6-deficient NPCs exhibit poor proliferation and a failure to differentiate into neurons and glia. Mechanistically, both JMJD3 and UTX are found to be enriched in gene loci essential for neural development in hNPCs, and KDM6 impairment leads to H3K27me3 accumulation and blockade of DNA accessibility at these genes. Interestingly, forced expression of neuron-specific chromatin remodelling BAF (nBAF) rescues the neuron/glia defect in KDM6-deficient NPCs despite H3K27me3 accumulation. Our findings uncover the differential requirement of KDM6s in specifying NPCs and neurons/glia and highlight the contribution of individual epigenetic regulators in fate decisions in a human development model.

[1] CAS Key Laboratory of Regenerative Biology, Joint School of Life Sciences, Guangzhou Institutes of Biomedicine and Health, Chinese Academy of Sciences, Guangzhou Medical University, Guangzhou 510530, China. [2] Nanfang Hospital, Southern Medical University, Guangzhou 510515, China. [3] University of Chinese Academy of Sciences, Beijing 100049, China. [4] Guangdong Provincial Key Laboratory of Stem Cell and Regenerative Medicine, South China Institute for Stem Cell Biology and Regenerative Medicine, Guangzhou Institutes of Biomedicine and Health, Chinese Academy of Sciences, Guangzhou 510530, China. [5] Guangzhou Regenerative Medicine and Health Guangdong Laboratory, Guangzhou 510005, China. [6] Shandong Medicinal Biotechnology Center, Shandong First Medical University & Shandong Academy of Medical Sciences, Jinan 250012, China. [7] Centre for Regenerative Medicine and Health, Hong Kong Institute of Science and Innovation, Chinese Academy of Sciences, Hong Kong, China. [8] These authors contributed equally: Yongli Shan, Yanqi Zhang, Yuan Zhao. *email: pan_guangjin@gibh.ac.cn

Neurogenesis plays a critical role in early human embryonic development and gives rise to uniquely sophisticated central and peripheral nervous systems that are very different from those of other species[1,2]. However, most of our knowledge about human neurogenesis depends on anatomical specimen analysis, while few molecular and cellular investigations can be attempted due to the lack of appropriate model systems. The prospect of in vitro neurogenesis modelling through stem cell-based modalities provides a unique opportunity to bridge the gap between our complex knowledge from anatomical observation and limited knowledge at the molecular and cellular levels[3–6].

In general, neurogenesis appears to be a sequential and highly ordered process that specifies cell lineages ranging from pluripotent stem cells (PSCs) to neural progenitor cells (NPCs) to various subtypes of neurons and glial cells[7–9]. During this process, specification of cell fate must be highly coordinated at all levels to ensure the generation of neurons and other related cell types of the human nervous systems. However, very little is known about how neural cell specification is regulated at the molecular level, although research in model organisms such as the fly, fish and mouse has shown that the orchestration between lineage-specific transcription factors (TFs) and epigenetic mechanisms may be crucial[10–16]. Specifically, neurogenesis starts when progenitor cells acquire DNA accessibility at loci essential for neurogenesis, which allows the binding of TFs to trigger lineage-specific programmes. Chromatin accessibility is mainly regulated through DNA and histone modifications including methylation, acetylation, ubiquitination, etc.[17]. These epigenetic modifications are believed to be important to specify the lineage fidelity of different cell types during neurogenesis[15,18–21]. For example, H3K27me3 is widely associated with important lineage genes and maintain them at a repressed but poised state in stem cells, as demonstrated by genome-wide mapping studies[22–25]. H3K27me3 is generated by polycomb repressive complex 2 (PRC2) and removed by KDM6 demethylases (comprising two members, JMJD3/KDM6B and UTX/KDM6A)[26,27], providing a quite precise way to control gene expression at the chromatin level during development. Given its repressive nature, H3K27me3 must be deployed to close chromatin loci unrelated to a particular cell fate and removed to open chromatin critical to that fate. This close-open logic was recently observed during the conversion of mouse embryonic fibroblasts (MEFs) to iPSCs[28,29]. The same logic might also apply to other cell fate decision processes, such as neurogenesis. Indeed, H3K27me3 has been implicated in neural development[30–32]. Furthermore, JMJD3 and UTX were shown to play important roles in adult neurogenesis in a mouse model[33,34]. However, how these epigenetic mechanisms precisely determine lineage specification and fate transition from human PSCs to neural progenitor cells (NPCs) and then subtypes of neural cells remains largely unknown.

Here, we knocked out KDM6s (JMJD3 and/or UTX), the known H3K27me3 demethylases, in human embryonic stem cells (hESCs) and showed that KDM6s (JMJD3 and/or UTX)-deficient human ESCs exit pluripotency and commit to NPC differentiation normally, but the resulting NPCs fail to transit into neurons and glia due to a lack of accessibility at loci essential for neurogenesis. Our findings reveal an essential role of this KDM6-dependent epigenetic mechanism in specifying NPC and neuron/glial lineage fidelity and fate decisions in human neurogenesis.

## Results

**KDM6s-deficient hESCs undergo normal neural differentiation.** Unlike those in all other species, human genetic mutants cannot be generated to test the function of a specific epigenetic modifier for H3K27me3, such as PRC2 and KDM6s, in vivo.

However, *UTX* mutations have been associated with Kabuki syndrome, a disease affecting 1 in 23000 children that causes underdeveloped intelligence[35,36]. In studies carried out in another species, mouse embryos with KDM6 deletion developed to full term and appeared to be normal at midgestation[37–39], thus raising questions regarding the role of H3K27me3 removal in fate decisions during embryonic development. To investigate the role of KDM6s in human neurogenesis, we deleted the catalytic domain of UTX and/or JMJD3[40] in H1 human ESCs, named H1-*UTX*$^{-/Y}$, H1-*JMJD3*$^{-/-}$ and H1-*JMJD3*$^{-/-}$/*UTX*$^{-/Y}$ (H1-dKO) (Supplementary Fig. 1a, b). Human ESCs in which UTX, JMJD3 or both were deleted maintained a normal karyotype and typical undifferentiated state in terms of morphology, pluripotent gene expression and cell cycle (Supplementary Fig. 1c–i), indicating that JMJD3 and UTX are dispensable in maintaining hESCs. We then triggered default neural differentiation in these cells under defined conditions through the dual inhibition of TGFb/BMP signalling[3,41] (Fig. 1a). Upon differentiation, all three KDM6 mutant hESC lines lacking UTX and/or JMJD3 exhibited a rosette-like morphology, the typical NPC phenotype in hESC differentiation (Fig. 1b). The pluripotent genes OCT4 and NANOG were fully suppressed, while the NPC genes PAX6, SOX2 and SOX1 were upregulated at day 16 of differentiation (Fig. 1c). As expected, UTX and/or JMJD3 expression was not detected in the corresponding knock-out cell lines during the whole differentiation process (Fig. 1c). These data indicate that the impairment of JMJD3 and/or UTX does not delay the exit of pluripotency and NPC differentiation in hESCs. Indeed, PAX6-positive cells and PAX6 protein levels were quite similar between wild-type (WT) cells and three KDM6 mutant hESC lines upon neural differentiation (Fig. 1d, e). Furthermore, immunostaining data showed that the rosette-like cells from WT cells and three mutant hESC lines highly expressed the typical NPC markers SOX2, NES (NESTIN), and PAX6 but not OCT4, a pluripotent marker (Fig. 1f). Together, these data demonstrate that JMJD3 and/or UTX deficiency in hESCs does not impede fate transition at the early stage of neural differentiation. Notably, the total levels of H3K27me3 and another histone modification, H3K4me3, were not significantly different between mutant and WT cells (Fig. 1g), indicating that the active removal of H3K27me3 by JMJD3 and UTX is not critical at the early stage of PSC neural differentiation.

**KDM6s are essential for the long-term proliferation of hNPCs.** To further characterize NPCs lacking UTX and/or JMJD3, we picked the rosette-like cells and maintained them as neural spheres in NPC medium (Fig. 2a). At early passages, they formed typical neural spheres similar to those of the wild-type cells (Fig. 2a). However, at later passages, neural sphere formations were reduced in the three mutant NPC lines, and the most severe defect was observed in H1-*JMJD3*$^{-/-}$/*UTX*$^{-/Y}$ (H1-dKO) NPCs (Fig. 2a). Consistently, the three mutant NPC lines at later passages showed substantially reduced proliferation and EdU incorporation (Fig. 2b, c, Supplementary Fig. 2a, b). Notably, these three KDM6 mutant NPC lines maintained an undifferentiated state at late passages, as the expression of NPC genes was not reduced in these cells (Fig. 2d). In addition, more apoptosis was detected in the three mutant NPC lines than in WT cells at late passages (Fig. 2e, f, Supplementary Fig. 2c). We then performed RNA-seq analysis of WT and H1-dKO NPCs at late passages (Fig. 2g–j). Consistent with their phenotype, upregulated genes in H1-dKO NPCs were enriched in functions related to apoptosis, while downregulated genes were related to NPC proliferation (Fig. 2h, i). The selected cell

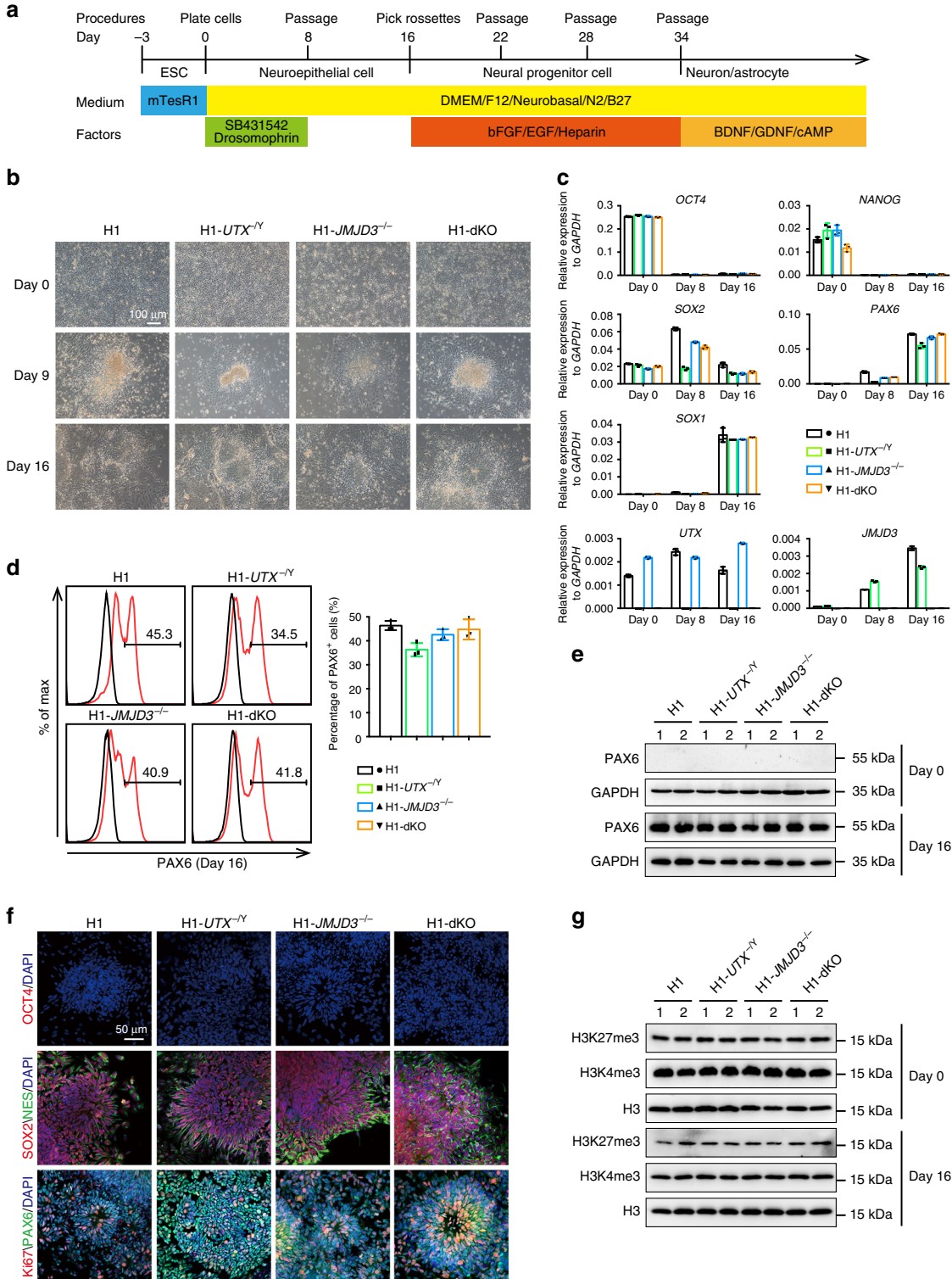

cycle genes were significantly reduced in H1-dKO NPCs (Fig. 2j). Together, these data indicate that the KDM6s JMJD3 and UTX are essential for long-term self-renewal and proliferation in human NPCs.

**KDM6s-deficient NPCs fail to generate neurons and astrocytes**. To further characterize the potential of the three KDM6 mutant NPC lines lacking UTX and/or JMJD3, we performed spontaneous differentiation by withdrawing growth factors that support

NPC self-renewal (Fig. 3a). After 28 days of culture in the absence of growth factors, WT NPCs (passage 2, P2) exited proliferation and differentiated into neurons and astrocytes, as detected by morphological changes and immunostaining for MAP2, a neuronal marker, and GFAP, an astrocyte marker (Fig. 3b–d). In contrast, it was difficult to detect neurons and astrocytes in all three KDM6 mutant NPC lines at the same point during their differentiation (Fig. 3b, c). This neuronal/glial defect was not due to cell apoptosis in differentiation, as the proportion of annexin

**Fig. 1 NPC differentiation of KDM6s-deficient hESCs. a** Overview of the default neural differentiation strategy for hESCs. hESCs maintained in mTeSR1 medium under monolayer conditions were treated with two SMAD inhibitors (5 μM SB431542/5 μM dorsomorphin) in the indicated defined medium. The rosette-like cells were picked at day 16 and expanded as neural spheres. For further differentiation, neural spheres were then plated on Matrigel and cultured in the indicated medium for spontaneous differentiation (see "Methods" sections for details.). hESCs, human embryonic stem cells. **b** Morphology of the wild-type (WT) H1 or KDM6-deficient hESC lines (H1-$UTX^{-/Y}$, H1-$JMJD3^{-/-}$, H1-dKO) under NPC differentiation conditions at day 0, day 9 or day 16. Scale bar, 100 μm. dKO, deletion of both $JMJD3$ and $UTX$. **c** qRT-PCR analysis of the expression of the pluripotent genes $OCT4/NANOG$ and the NPC markers $SOX2/PAX6/SOX1$ and $UTX/JMJD3$ at day 0, day 8 and day 16 of neural differentiation. Wild-type H1 hESCs served as controls. The data represent the mean ± SD (standard deviation) from three independent replicates ($n = 3$). **d** FACS analysis of PAX6$^+$ cells at day 16 of neural differentiation in the indicated cells. The data represent the mean ± SD from three independent replicates ($n = 3$). **e** Western blot of PAX6 proteins in the indicated cells at day 0 or day 16 of neural differentiation. **f** Immunostaining for the pluripotent marker OCT4, the NPC markers SOX2/NES/PAX6, and the proliferation marker Ki67 in WT NPCs or three KDM6 mutant NPC lines lacking UTX, JMJD3 or both. Scale bar, 50 μm. **g** Western blot of H3K27me3 and H3K4me3 in the indicated cells at day 0 or day 16 of neural differentiation. All error bars throughout the figure represent the SD (standard deviation) from three independent replicates ($n = 3$). Source data are provided as a Source Data file. See Supplementary Fig. 1 for more detail. NPC neural progenitor cell, WT wild type.

V$^+$ apoptotic cells among both WT cells and all three mutant cell lines was not significant (Supplementary Fig. 3a). Moreover, these three KDM6 mutant NPC lines maintained a rosette-like morphology and expressed the NPC genes $SOX2$ and $NES$ at day 28 of spontaneous differentiation (Fig. 3b, d). qRT-PCR analysis further confirmed that the NPC genes were highly expressed, while the neuronal and astrocyte genes remained repressed in the three KDM6 mutant cell lines at day 28 of differentiation (Fig. 3e). We then generated whole-genome transcriptome data from undifferentiated or differentiated WT and $JMJD3^{-/-}/UTX^{-/Y}$ (dKO) NPCs (Fig. 3f). Consistent with the phenotype shown above, Spearman's rank correlation analysis clearly showed the much closer relationships between WT NPCs, dKO NPCs and differentiated dKO NPCs at day 28 (dKO-D28) compared with the differentiated WT NPCs at day 28 (WT-D28) (Fig. 3f). We further showed that the differentiation defect in dKO NPCs could be rescued by the re-expression of exogenous $JMJD3/UTX$ in dKO hESCs (Supplementary Fig. 3b, c), demonstrating that the phenotype is specific to KDM6s. Together, these data demonstrate that the KDM6s JMJD3 and UTX are required for the fate transition of NPCs into neurons and astrocytes in human neurogenesis.

**KDM6s deficiency reduces chromatin accessibility in hNPCs.** To investigate the molecular mechanisms underlying the differentiation defect in NPCs lacking UTX and JMJD3, we performed whole-genome transcriptome and transposase-accessible chromatin using sequencing (ATAC-seq) analyses in WT NPCs and dKO NPCs at early passage (passage 2). Genes upregulated in dKO NPCs were related to extracellular matrix organization, angiogenesis and skeletal system development (Fig. 4a, b), indicating that the alternative lineage genes lost their full repression in the absence of JMJD3 and UTX. In contrast, the downregulated genes in dKO NPCs were almost exclusively related to neural progenitor cell proliferation and neural development (Fig. 4c). These data indicate that JMJD3 and UTX are essential, as they specify the lineage fidelity of human NPCs through repressing alternative lineage genes while maintaining neural lineage genes.

By ATAC-seq analysis, we identified a significant number of differentially accessible chromatin regions between WT NPCs and dKO NPCs (Fig. 4d, e). The open chromatin associated with the promoter regions of approximately one thousand genes (TSS ± 3 kb) in WT NPCs became more closed in dKO cells (Fig. 4e). Again, these genes in which chromatin was closed in dKO NPCs were almost exclusively related to neural development (Fig. 4d). For example, many known critical NPC regulators showed much more closed chromatin in dKO NPCs, including $SOX2$, $SOX1$, $NES$, $FOXG1$ and others (Fig. 4g). In addition, NOTCH pathway genes[42,43] that were shown to regulate the self-renewal and

differentiation of NPCs in a mouse model also showed substantially decreased chromatin accessibility in dKO NPCs (Fig. 4g), which might explain the proliferation defect of the mutant hNPC lines. Notably, the neuron-specific genes $MAP2$, $TUBB3$ (beta III Tubulin), $NERUOG1$, $NEUROG2$, $NEUROD1$, etc., also showed open chromatin in WT hNPCs (Fig. 4g), indicating that these neural lineage subtype genes are predisposed towards rapid expression in NPCs. Again, these genes also showed reduced chromatin accessibility in dKO NPCs (Fig. 4g), which might be the reason for their failure to differentiate into neurons and glia. A small number of genes related to extracellular matrix functions, such as some collagen genes, displayed a slight increase in chromatin accessibility in dKO NPCs (Fig. 4d, g). These changes in chromatin state were highly correlated with gene transcription, as the transcriptionally downregulated genes in dKO NPCs showed substantially reduced chromatin accessibility compared with the upregulated genes (Fig. 4a, f).

We further examined the binding motifs of known transcription factors (TFs) enriched in differentially accessible chromatin regions between WT and dKO hNPCs. The closed chromatin regions in dKO cells were highly enriched in the motifs of essential neural regulators with $P$ values indicating extreme significance (Fig. 4h). All the top ten TF motifs most enriched in dKO-closed regions have been reported to be essential neural regulators, for example, RFX2 ($P = 1e^{-1064}$), SOX2 ($P = 1e^{-661}$), and OTX2 ($P = 1e^{-403}$)[41,44,45] (Fig. 4h). In contrast, enriched motifs in dKO-opened regions were related to cellular process regulations, including the AP1 family factors FRA1, FOSL2, ATF3, etc.[46] (Fig. 4l). Notably, AP1 family factors are known to regulate the cell cycle and apoptosis, indicating that the opening of these regions for AP1 binding might contribute to the proliferation defect in dKO cells[47,48]. Taken together, these data demonstrate that the KDM6s JMJD3 and UTX are essential to promote a permissive chromatin state defining lineage fidelity in human NPCs.

**KDM6s promote epigenetic switching in hNPC differentiation.** To further investigate the role of KDM6s in NPC differentiation, we also generated ATAC-seq and transcriptome data in differentiated WT or dKO cells on day 28 (WT-D28 or dKO-D28, respectively). The upregulated genes in WT-D28 cells (mainly neurons, Fig. 3c) were clearly highly related to neuronal regulatory process, such as axon development and synapse function. (Fig. 5a, b). The downregulated genes were almost all related to mitosis and cell cycle regulation processes, such as DNA replication, chromosome segregation, and G1/S transition (Fig. 5a, c), indicating cell cycle exit during NPC-to-neuron transition. In contrast, these up- or downregulated genes in WT-D28 cells were aberrantly regulated in differentiated dKO-D28 cells (Fig. 5a).

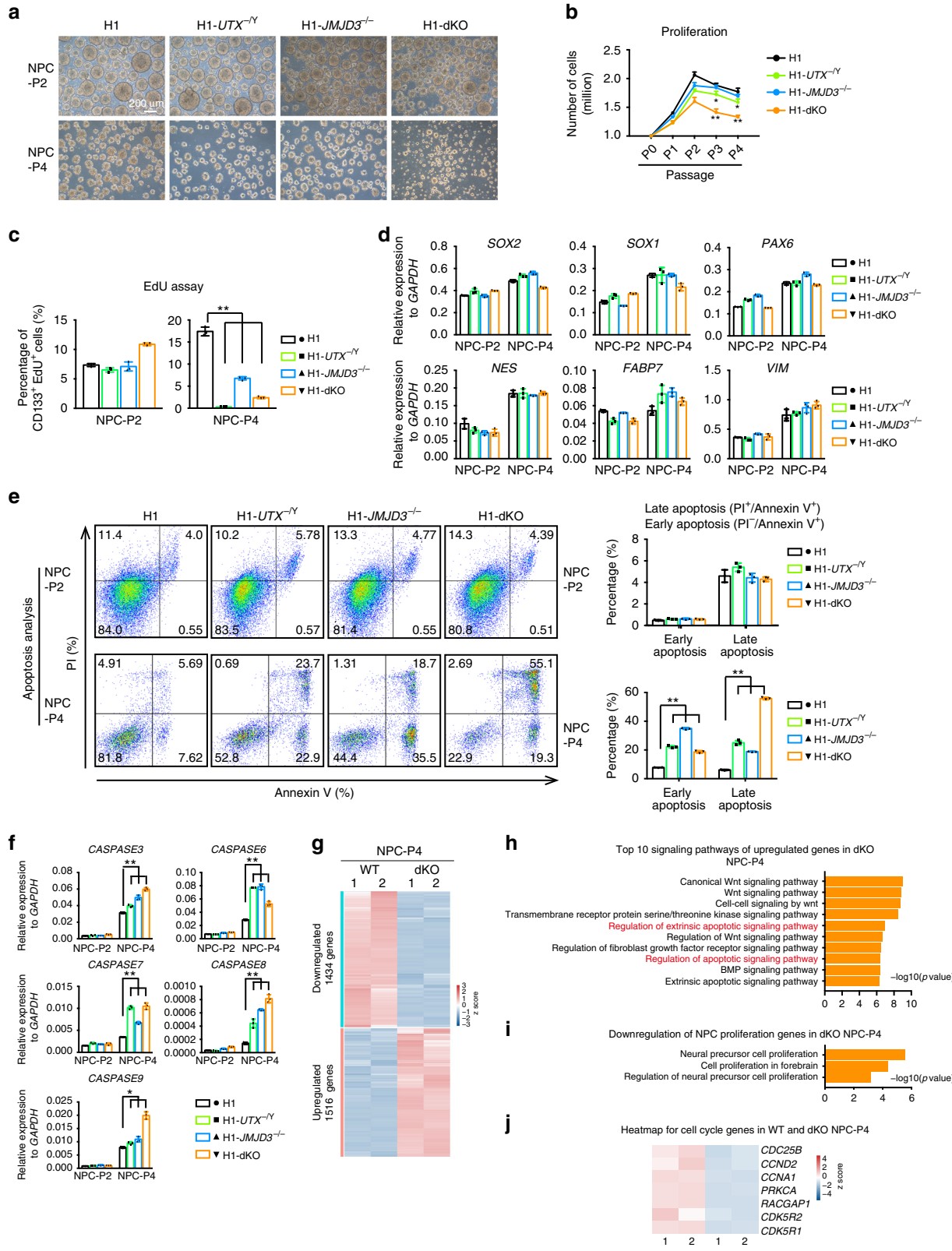

In terms of the chromatin state, we identified massive chromatin regions that showed reduced accessibility upon the differentiation in WT NPCs (Fig. 5d). These genes with closed chromatin upon differentiation were shown to be related to the regulation of cellular processes, such as protein synthesis, RNA biosynthesis, DNA replication, and the cell cycle. (Fig. 5e), consistent with the inactive cellular process observed in post-mitotic neurons. Notably, only a very small number of genes (10) showed increased chromatin accessibility upon differentiation (Fig. 5d), indicating that the NPC-to-neuron/glia transition involves mainly epigenetic closing rather than opening. As expected, the transcriptionally downregulated genes in the

**Fig. 2 KDM6s are essential for proliferation in human NPCs. a** Morphology of indicated NPCs maintained as neural spheres at passage 2 (P2) or passage 4 (P4). Scale bar, 200 μm. **b** Proliferation curve of the indicated NPCs at different passages. Significance was determined using unpaired two-tailed Student's t-tests. *P < 0.05. **P < 0.01. The data represent the mean ± SD (standard deviation) from three independent replicates (n = 3). **c** EdU incorporation assay of the indicated CD133+ NPCs at passage 2 (P2) or passage 4 (P4) (see "Methods" section for details.). Significance was determined using unpaired two-tailed Student's t-tests. **P < 0.01. The data represent the mean ± SD from three independent replicates (n = 3). **d** qRT-PCR analysis on the expression of the NPC markers *SOX2*, *SOX1*, *PAX6*, *NES*, *FABP7*, and *VIM* in the indicated NPCs at passage 2 (P2) or passage 4 (P4). The data represent the mean ± SD from three independent replicates (n = 3). **e** Apoptosis assay in the indicated cells at passage 2 (P2) or passage 4 (P4). PI- and/or annexin V-positive cells were analysed by FACS. The significance level was determined using unpaired two-tailed Student's t-tests. **P < 0.01. Error bars represent the mean ± SD from three independent experiments (n = 3). **f** qRT-PCR analysis of the expression of the apoptosis markers *CASPASE3*, *CASPASE6*, *CASPASE7*, *CASPASE8*, and *CASPASE9* in the indicated NPCs at passage 2 (P2) or passage 4 (P4). The significance level was determined using unpaired two-tailed Student's t-tests. *P < 0.05. **P < 0.01. The data represent the mean ± SD from three independent replicates (n = 3). **g** Heatmap of up- or downregulated genes in dKO NPCs and WT NPCs at passage 4 (P4). dKO, deletion of both JMJD3 and UTX. **h** Top 10 GO terms for signalling pathways enriched in genes upregulated in dKO NPCs at passage 4 (P4). **i** GO terms enriched in genes downregulated in dKO NPCs at passage 4 (P4). **j** Heatmap of cell cycle-related genes in WT and dKO NPCs at passage 4. All error bars throughout the figure represent the SD (standard deviation) from three independent replicates (n = 3). Source data are provided as a Source Data file. See Supplementary Fig. 2 for more detail. WT wild type.

NPC-to-neuron/glia transition exhibited the most significantly reduced chromatin accessibility (Fig. 5a, f).

We then analysed the whole-genome transcriptome and global chromatin accessibility in differentiated dKO-D28 NPCs. Consistent with the differentiation defect phenotype, genes related to neuron regulation were not upregulated in dKO-D28 cells (Fig. 5g). In contrast, the cell cycle- and proliferation-related genes that were expected to be repressed were well maintained (Fig. 5g). Generally, the chromatin accessibility of many genes essential for neurogenesis, such as *NEUROG1/2* and *NEUROD1/4*, was not increased in dKO-D28 cells. (Fig. 5h, i). Taken together, our findings reveal that the KDM6s JMJD3 and UTX are essential for the epigenetic switch in fate transition from human NPCs to neural subtypes.

**KDM6s knock-out results in H3K27me3 accumulation in neurogenesis**. Because KDM6s are H3K27me3 demethylases, we next examined the global level of H3K27me3 in WT and dKO cells through western blotting (Fig. 6a). The total level of H3K27me3 gradually increased from hESCs to NPCs and then to differentiated neural cells (Fig. 6a). Upon the deletion of JMJD3 and UTX, significant H3K27me3 accumulation was observed in differentiated cells at D28 but not in cells at the NPC stage (Fig. 6a), indicating that H3K27me3 demethylation by KDM6s is dominant at a later stage of neural differentiation rather than at the NPC stage. We then performed ChIP-seq for the genome-wide mapping of H3K27me3 in WT and dKO NPCs and their differentiated cells (Fig. 6b, Supplementary Fig. 4a, b). The overall H3K27me3 intensity was much higher in dKO cells than in WT cells (Fig. 6b). Upon differentiation, a significant number of H3K27me3-associated genes in WT NPCs lost H3K27me3 in differentiated cells at D28 (Fig. 6c, upper left panel). These genes were almost exclusively related to neuron regulation (Fig. 6c, upper right panel). However, H3K27me3 was highly maintained on these genes in differentiated dKO-D28 cells (Fig. 6c, upper middle panel). As expected, these genes were transcriptionally upregulated in differentiated WT neural cells but maintained at lower expression levels in dKO-D28 cells (Supplementary Fig. 4c–e). These data suggest that the activation of genes essential for neuron specification relies mainly on active H3K27me3 demethylation by KDM6s. On the other hand, we also detected increased H3K27me3 in a group of genes, and expression of these genes was more suppressed in differentiated WT neurons (Fig. 6c, lower left panel; Supplementary Fig. 6a, right panel). These genes were found to be related to the regulation of other cell lineages, such as immune cells (Fig. 6c, lower right panel).

These data indicate that KDM6s might selectively bind essential neural genes in NPCs to define lineage fidelity. To confirm this, we sought to perform ChIP-seq to investigate the whole genome-binding sites of JMJD3 and UTX in human NPCs. Because we failed to find good antibodies against JMJD3 or UTX for ChIP-seq, we knocked in 3 × FLAG at the C-termini of UTX and JMJD3 in hESCs and named these cell lines H1-*UTX*-KI-3 × FLAG and H1-*JMJD3*-KI-3 × FLAG, respectively. Double-positive clones were successfully expanded without changes in the undifferentiated state of hESCs. FLAG-tagged UTX and JMJD3 were clearly detected in these clones through western blotting (Fig. 6d). UTX- or JMJD3-bound genes in hNPCs were then identified through ChIP-seq using anti-FLAG antibody (Fig. 6d). Interestingly, almost all UTX-bound promoters showed detectable JMJD3 co-localization (Fig. 6d). These genes bound to UTX/JMJD3 in hNPCs were extremely enriched in neuron development functions and cell cycle regulation[49] (Fig. 6d), demonstrating that UTX/JMJD3 directly co-bind essential neuron genes in hNPCs to ensure their subsequent differentiation. Indeed, the overall H3K27me3 intensities were significantly reduced, and transcriptional levels of UTX- or JMJD3-bound genes, but not un-bound genes, were upregulated in both hNPCs and differentiated cells at D28 (Fig. 6e, f, Supplementary Fig. 4f). Accordingly, these UTX- or JMJD3-bound genes also showed more accessible chromatin than the un-bound genes (Fig. 6e, right panel). Notably, we detected a set of genes showing high JMJD3 binding but low or no UTX binding in human NPCs; these genes are related to autophagy (Fig. 6d), indicating that JMJD3 might play additional roles in NPCs. Nonetheless, taken together, these data demonstrate that KDM6s-dependent H3K27me3 demethylation is essential in specifying neural subtypes from NPCs through direct binding to neural lineage-specific genes.

**KDM6s are required for the expression of nBAF genes**. The BAF (Brg/Brm-associated factor) complex, which functions in ATP-dependent SWI/SNF chromatin remodelling, was shown to be essential in neural development in a mouse model[50–52]. Specification of post-mitotic neurons accompanied the exchange of subunits within the BAF complex from NPC-specific BAF subunits (npBAF) to neuron-specific ones (nBAF)[52,53]. We then examined the expression of BAF subunit genes in dKO NPCs and their differentiated cells in more detail. mRNA levels of the core BAF subunits BRG1 and BRM were similar in WT and dKO NPCs and their differentiated cells (Fig. 7a, upper panel). However, their protein levels were substantially reduced in differentiated dKO-D28 cells (Fig. 7a, middle panel), indicating that BRG1 and BRM undergo degradation in differentiated dKO cells. Treatment with the proteasome inhibitor MG132 largely rescued BRG1 and BRM protein degradation in dKO cells (Fig. 7a, lower panel), indicating that

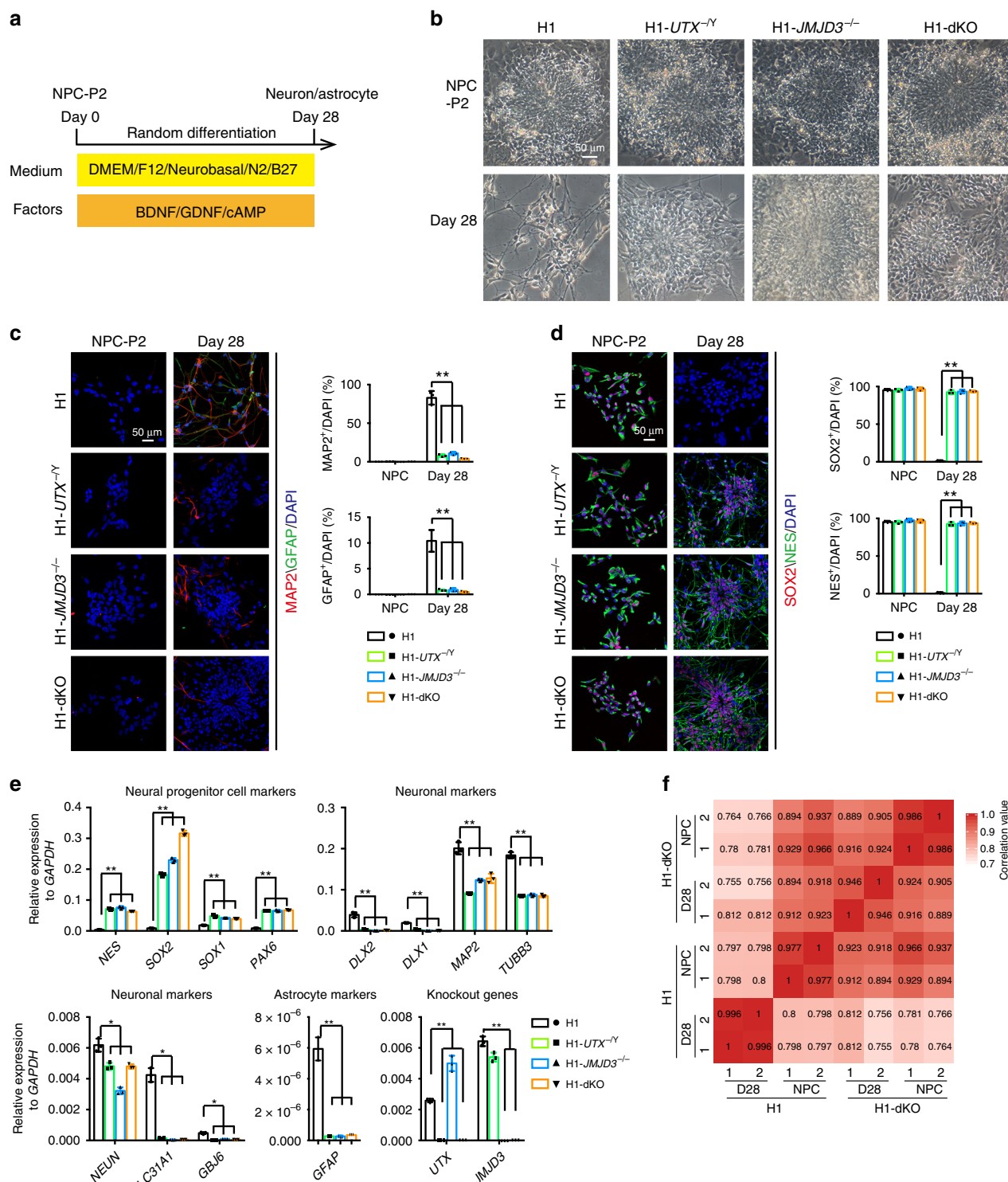

KDM6s might promote the protein stability of core BAF subunits. Indeed, co-expression of JMJD3 and UTX significantly inhibited the degradation of BRG1 or BRM (Supplementary Fig. 5a). Among npBAF genes, *BAF53A* showed no significant difference in expression between undifferentiated WT and dKO NPCs (Fig. 7b). However, upon differentiation, while *BAF53A* was downregulated in WT cells, it remained highly expressed in differentiated dKO cells (Fig. 7b). On the other hand, upon neuron differentiation, the nBAF gene *BAF53B* was significantly upregulated in WT cells but

repressed in dKO cells (Fig. 7c). These data indicate that KDM6-mediated H3K27me3 demethylation is required for nBAF expression. Consistently, while H3K27me3 on the core BAF genes and npBAF genes in undifferentiated NPCs was not obvious, it was highly enriched in nBAF genes (Fig. 7d). Upon differentiation to neurons, H3K27me3 on nBAF genes was eliminated in WT cells but maintained in differentiated dKO cells (Fig. 7d). These data demonstrate that JMJD3 and UTX are essential for BAF subunit exchange in neurogenesis.

**Fig. 3 KDM6s-deficient NPCs fail to differentiate into neurons and glia. a** Strategic diagram of the spontaneous differentiation of human NPCs. Wild-type (WT) NPCs or three KDM6 mutant NPC lines lacking UTX, JMJD3 or both at passage 2 (NPC-P2) were plated in differentiation media lacking FGF2 and EGF and cultured for 28 days for differentiation. **b** Morphology of the undifferentiated WT NPCs or three KDM6 mutant NPC lines and their differentiated cells at day 28. Scale bar, 50 μm. **c** Immunostaining for the neuronal marker MAP2 and the glial marker GFAP in the indicted undifferentiated or differentiated NPCs. Scale bar, 50 μm. Quantity data from MAP2$^+$ or GFAP$^+$ cells were analysed. Significance level was determined by unpaired two-tailed Student's $t$-tests. $**P$ < 0.01. The data represent the mean ± SD (standard deviation) from three independent replicates ($n = 3$). **d** Immunostaining for the NPC markers SOX2/ NES in the indicated undifferentiated or differentiated NPCs. Scale bar, 50 μm. The percentage of SOX2$^+$ or NES$^+$ cells was analysed. Significance was determined by unpaired two-tailed Student's $t$-tests. $**P < 0.01$. The data represent the mean ± SD from three independent replicates ($n = 3$). **e** qRT-PCR analysis of the expression of the NPC markers *NES/SOX2/SOX1/PAX6*, the neuronal markers *DLX2/DLX1/MAP2/TUBB3/NEUN/SCL31A1 /GBJ6*, the glial marker *GFAP*, and *UTX/JMJD3* in the indicated differentiated cells at day 28. The significance level was determined using unpaired two-tailed Student's $t$-tests. $**P < 0.01$. The data represent the mean ± SD from three independent replicates ($n = 3$). **f** Spearman's rank correlation analysis of the whole-genome transcriptome of the indicated undifferentiated or differentiated NPCs. All error bars throughout the figure represent the SD (standard deviation) from three independent replicates ($n = 3$). Source data are provided as a Source Data file. See Supplementary Fig. 3 for more information.

**BAF53B rescues the neuronal/glial defect in JMJD3/UTX-dKO NPCs.** We then sought to examine whether nBAF could rescue the differentiation defect in dKO cells. Lentivirus expressing BAF53B, the neuronal BAF subunit, was transduced into dKO NPCs (Fig. 7e). Upon spontaneous differentiation, BAF53B rescued the neuronal/glial differentiation defect in dKO cells, as evidenced by changes in morphology and marker expression (Fig. 7e). These data demonstrate that nBAF can drive fate transition to neurons independent of JMJD3 and UTX. Indeed, the accumulation of H3K27me3 in dKO cells was not reduced with the forced expression of *BAF53B* (Fig. 7f). Interestingly, based on the results of ATAC-seq, the chromatin regions closed by KDM6 dKO were re-opened in BAF53B-rescued cells (Fig. 7g). Similarly, the opening of chromatin regions by JMJD3/UTX dKO reduced their accessibility in BAF53B-rescued cells (Fig. 7g). Together, these data reveal that nBAF can drive neuron fate transition independent of KDM6s-dependent H3K27me3 demethylation.

## Discussion

The central nervous system (CNS) comprises an enormous array of diversified cell types developed from neural progenitor/stem cells via a highly orchestrated process. The molecular mechanisms driving the precise fate transition from NPCs to neural subtypes remain poorly understood. Our studies reveal the previously unknown but essential role of KDM6s-dependent H3K27me3 demethylation in specifying lineage fidelity and fate decision in human neurogenesis. Human ESC-derived NPCs without the KDM6s JMJD3 and/or UTX displayed reduced self-renewal and a failure to undergo fate transition into post-mitotic neurons and astrocytes. These KDM6s were shown to be essential to define a permissive chromatin state in NPCs and ensure epigenetic switching during their differentiation. In the absence of JMJD3 and UTX, H3K27me3 accumulated on genome loci essential for neurogenesis and blocked chromatin accessibility. Interestingly, both KDM6 members, JMJD3 and UTX, are required in this process, indicating that they are not functionally redundant in this case. Many factors that were identified in mice as critical neural regulators, for example, NOTCH signalling and BAF complex genes, are regulated by JMJD3 and UTX in human NPCs (Figs. 4 and 6). Moreover, KDM6s-mediated H3K27me3 modulation was also shown to be essential in the epigenetic switch during the transition from proliferating NPCs to post-mitotic neurons and astrocytes. In the absence of JMJD3 and UTX, DNA replication- and cell cycle-related genes remained highly expressed during NPC differentiation (Fig. 5). This deficiency might be due to the global inaccessibility of chromatin regions enriched in the motifs of those key neural regulators. These neural regulators include the BAF complex, which has been shown to be essential in neural development. Our data show that

BAF subunit genes are differentially regulated by KDM6s during fate transition from PSCs to NPCs and then to neurons (Fig. 7). In the absence of JMJD3 and UTX, npBAF genes are normally expressed in the fate transition from PSCs to NPCs. However, during further differentiation to neural cell subtypes, JMJD3 and UTX are clearly required in the exchange of npBAF to nBAF. Indeed, forced expression of BAF53B, the neuron-specific BAF, fully rescued the neuronal defect in human JMJD3 and UTX dKO NPCs (Fig. 7f). Interestingly, BAF53B re-set a chromatin state permissive to neuronal/glial differentiation in JMJD3 and UTX dKO NPCs (Fig. 7g). In addition, knockdown of npBAF in dKO NPCs promoted their subsequent neuronal/glial differentiation, although overexpression of npBAF failed to rescue their proliferation defect (Supplementary Fig. 5b–d). On the other hand, a previous report showed that the BAF units 155 and 170 interacted with UTX/JMJD3 to promote their demethylase function in an adult neurogenesis mouse model[54]. Our data showed that the core BAF units underwent degradation in differentiated neural cells in the absence of JMJD3 and UTX (Fig. 7a, Supplementary Fig. 5a), indicating that this interaction is also essential for BAF complex stability. Together, these findings reveal a close inter-dependent mechanism between JMJD3/UTX and BAF subunits in regulating fate transition during neural development.

PRC2-mediated H3K27me3 is a well-recognized epigenetic mechanism involved in the repression of lineage genes determined by the normal development of various cell lineages[25,55–59]. In human pluripotent stem cells, PRC2 and H3K27me3 bind almost all the early-lineage genes and are supposed to generally maintain them at a repressed state poised towards rapid activation upon differentiation[22,23,57]. However, our recent report showed that deletion of PRC2 in hPSCs preferentially led to spontaneous meso-endoderm differentiation, rather than the expected differentiation into all three germ layer lineages[30]. Thus, despite their broad genome association with lineage genes, the role of epigenetic regulators of H3K27me3 in fate decision appears to be specific to humans. Interestingly, PRC2 is clearly required to specify NPCs from human PSCs[30]. In contrast, KDM6s, the H3K27me3 demethylases, seem to be less critical during fate transition from human PSCs to NPCs (Fig. 1). This interesting phenomenon suggests that the methylation rather than de-methylation of H3K27me3 is important at the early stage of human neural development. However, KDM6s are absolutely required at later stages of neurogenesis during the transition from NPCs to neural subtypes (Fig. 3). Our findings reveal the differential requirement of the methylation and de-methylation of H3K27me3 during sequential lineage specification from pluripotency to neural subtypes in human neurogenesis, which has not been previously recognized. Indeed, deletion of *UTX* in another hESC line, HN10 cells[60], resulted in exactly the same

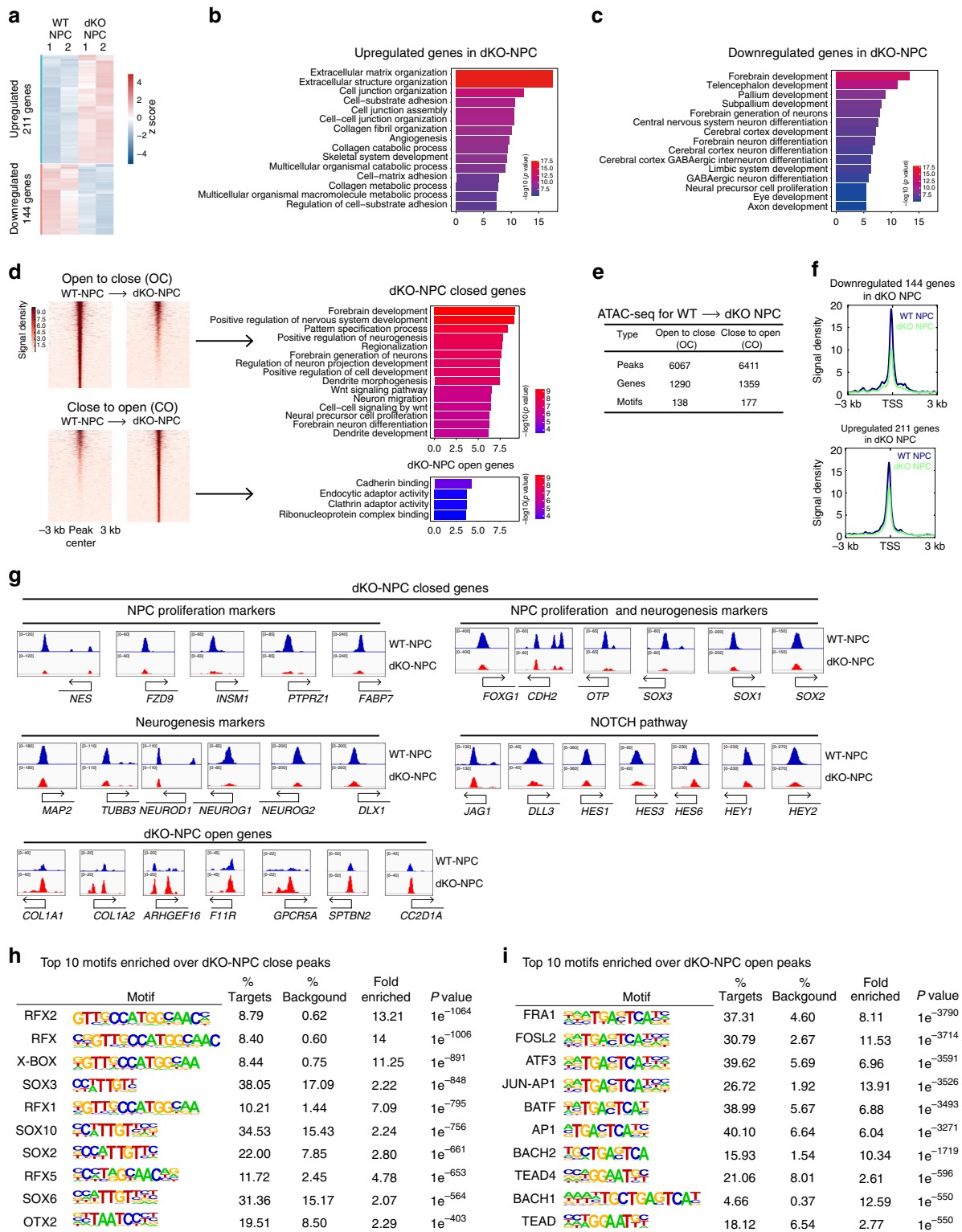

**Fig. 4 KDM6s deficiency decreases chromatin accessibility in hNPCs. a** Heatmap of up- or downregulated genes in dKO NPCs and wild-type (WT) NPCs at passage 2. **b, c** Top 15 GO terms most enriched in genes upregulated or downregulated in dKO NPCs. **d** Left panel, heatmap of ATAC-seq data indicating differentially accessible chromatin regions between WT NPCs and dKO NPCs. Right panel, top 15 GO terms most enriched in genes associated with decreased and increased chromatin accessibility in dKO NPCs. OC, open-to-closed, indicates regions with reduced chromatin accessibility in dKO NPCs. CO, closed-to-open, indicates regions with increased chromatin accessibility in dKO NPCs. **e** The numbers of OC or CO regions described in **d**. **f** ATAC-seq analysis of downregulated and upregulated genes in dKO NPCs and WT NPCs. **g** Genomic views of selected genes in WT NPCs or dKO NPCs from ATAC-seq data. These selected genes include those with functions of NPC proliferation, neuron development markers, NOTCH pathway genes and genes with increased chromatin accessibility in dKO NPCs. **h, i** Top 10 motifs most enriched over the OC or CO regions described in **d**.

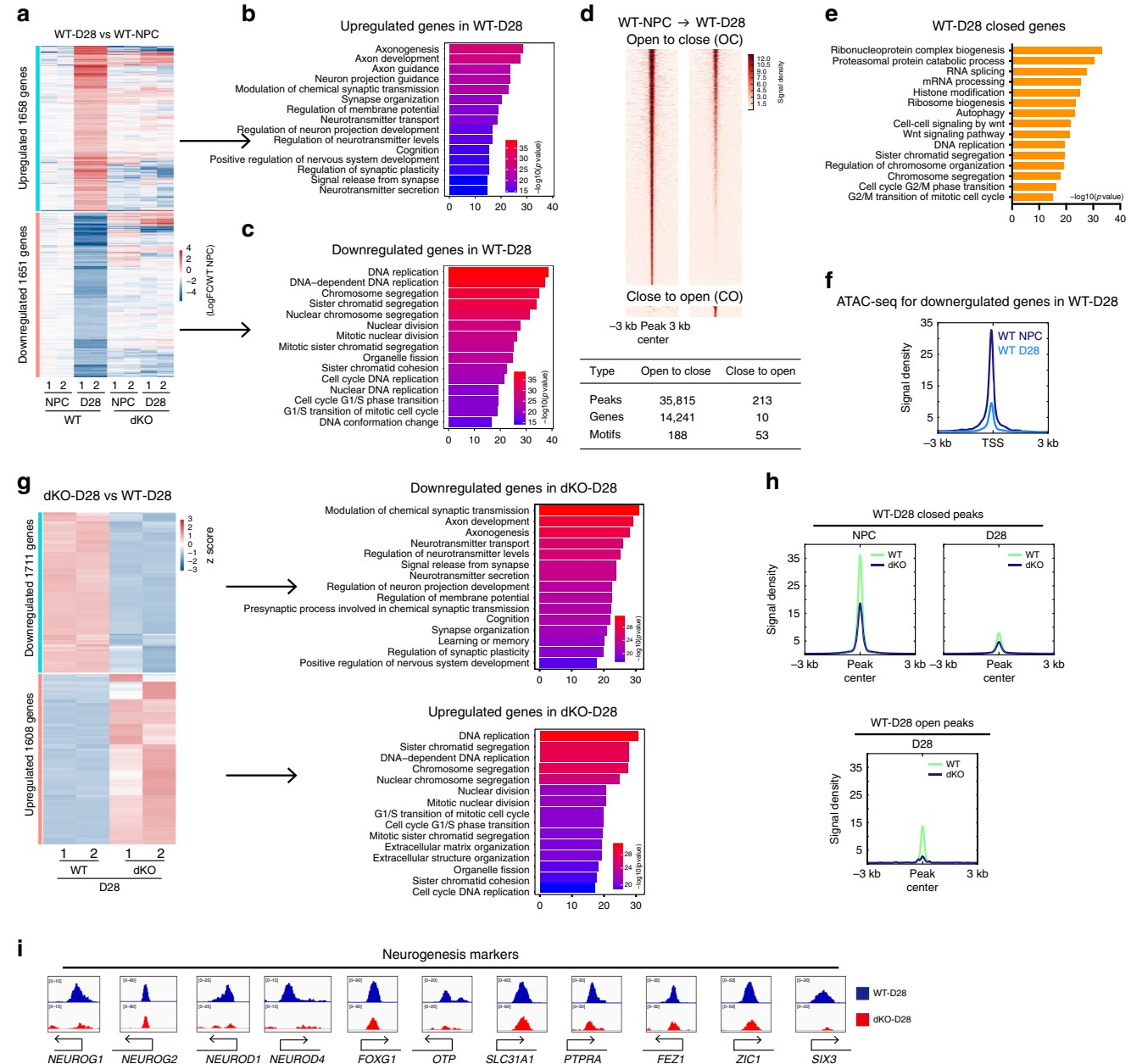

**Fig. 5 KDM6s are essential in the epigenetic switch from NPCs to neural subtypes. a** Heatmap of upregulated or downregulated genes in wild-type (WT)-D28 compared with wild-type (WT) NPCs. Expression levels are shown as fold changes (log2) compared with expression levels in WT NPCs. **b**, **c** Top 15 GO terms most enriched in up- or downregulated genes in differentiated WT-D28 cells. **d** Left panel, heatmap of ATAC-seq data indicating regions with differential chromatin accessibility between WT NPCs and differentiated cells at D28. Right panel, numbers of OC or CO regions. OC, open-to-closed, indicates regions with reduced chromatin accessibility in differentiated cells at D28. CO, closed-to-open, indicates regions with increased chromatin accessibility in differentiated cells at D28. **e** GO analysis for OC region-associated genes described in **d**. **f** ATAC-seq analysis indicating genes downregulated in differentiated WT-D28 cells compared with WT NPCs. **g** Heatmap and GO analysis of differentially expressed genes between WT and dKO cells at day 28 of differentiation. dKO, deletion of both JMJD3 and UTX. **h** Signal densities from ATAC-seq data for the indicated regions described in **d**. **i** Genomic views of genes involved in neuronal development indicated by ATAC-seq data from differentiated WT-D28 or dKO-D28 cells. WT wild type.

phenotype in neural differentiation (Supplementary Fig. 6), further highlighting the essential and universal role of KDM6 in human neurogenesis.

## Methods

**Cell culture**. The human embryonic stem cell lines H1 (Wi Cell) and HN10 (kindly provided by our collaborative colleagues in Hainan Provincial Key Laboratory for Human Reproductive Medicine and Genetic Research, generated with a published method[60]) and their knock-out cell lines (H1-$UTX^{-/Y}$,

H1-$JMJD3^{-/-}$, H1-$JMJD3^{-/-}/UTX^{-/Y}$ (H1-dKO), HN10-$UTX^{-/Y}$) were maintained on Matrigel (Corning)-coated plates in mTeSR1 (STEMCELL Technologies). These hES cells were passaged every 3 days, and the culture medium was replaced with fresh culture medium every day. These cells were maintained at 5% $CO_2$.

**Gene knock-out and knock-in in human ESCs**. We designed guide RNAs (gRNAs) for the knock-out and knock-in of *UTX*, *JMJD3* on the website crispr.mit.edu[61]. We used pX330 (Addgene) containing the Cas9 protein sequence and gRNA. For *UTX* and *JMJD3* knock-out, homologous arm donor DNAs targeting

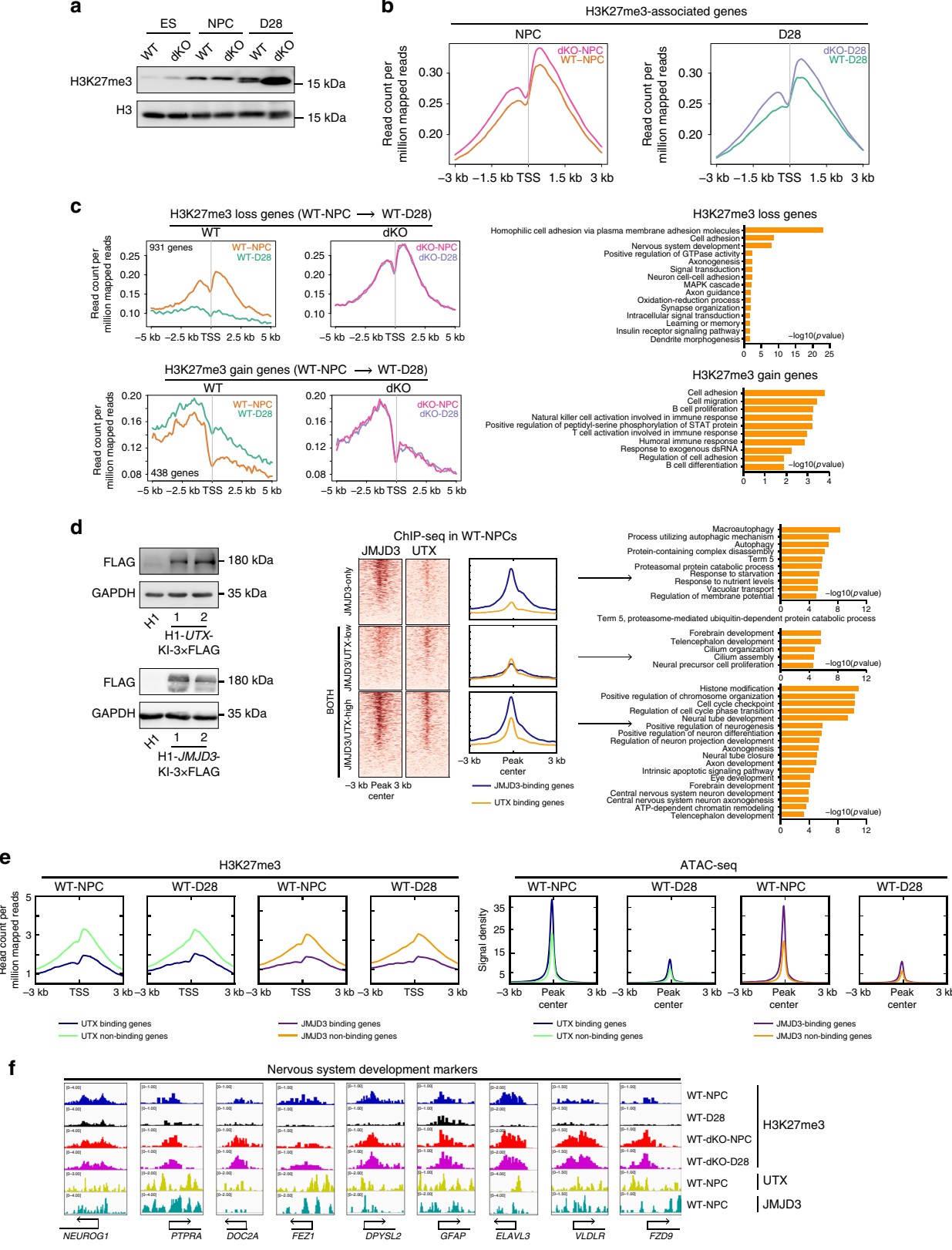

these genes containing left and right homology arms and a LoxP-flanked PGK-puromycin cassette were used. For UTX-3 × FLAG and JMJD3-3 × FLAG knock-in, the left homologous arm contained a 3 × FLAG tag. To target each gene, $1 \times 10^6$ hESCs were electroporated with 2 μg of donor DNA and 4 μg of pX330 plasmid containing the Cas9 protein sequence and the corresponding gRNAs for the knock-out and knock-in of UTX or JMJD3. Then, these electroporated cells were plated on

six-well plates coated with Matrigel in mTeSR1 medium with 10 μM Y-27632 (Sigma) for 1 day. Puromycin (1 μg mL$^{-1}$, Gibco) was used to select positive clones. To obtain UTX and JMJD3 double knock-out (dKO) cells, the antibiotic cassette in H1-JMJD3$^{-/-}$ was deleted using the Cre-LoxP system. H1-JMJD3$^{-/-}$ hESCs ($1 \times 10^6$) were electroporated with 0.4 mg of Cre mRNA. Then, 500 electroporated cells were plated on six-well plates coated with Matrigel in mTeSR1

**Fig. 6 KDM6s deficiency in human NPCs leads to H3K27me3 accumulation. a** Western blot for H3K27me3 in the indicated cells. **b** Signal densities of H3K27me3-associated genes in the indicated wild-type (WT) or dKO cells from ChIP-seq data. dKO, deletion of both JMJD3 and UTX. **c** Left panel, signal densities for H3K27me3 in genes that showed decreased and increased (upper and bottom panel) H3K27me3 modification in differentiated wild-type cells at day 28. Right panel, GO terms enriched in genes with decreased or increased H3K27me3 in differentiated wild-type cells at day 28 described in the left panel. **d** Left panel, western blot for FLAG in the *UTX*-3 × FLAG and *JMJD3*-3 × FLAG knock-in hES cells. Middle, signal densities and heatmap of ChIP-seq data from UTX- and JMJD3-associated genes in wild-type (WT) NPCs. Right panel, GO analysis of JMJD3- and both UTX/JMJD3-binding genes in NPCs. **e** H3K27me3 enrichment (left panel) and ATAC-seq (right panel) analysis of UTX- or JMJD3-bound or un-bound genes in the indicated wild-type (WT) NPCs or differentiated neural cells at D28. **f** Genomic views of the H3K27me3 modification and UTX and JMJD3 binding of genes related to neural development in the indicated cell lines. More information is also provided in Supplementary Fig. 4.

medium with 10 μM Y-27632 (Sigma) for 2 days. Then, *UTX* was deleted from H1-*JMJD3*$^{-/-}$ hESCs without the antibiotic cassette following a previous procedure.

For gene knock-out, 30–50 ng of genomic DNA from the knock-out cell clones extracted with a TIANamp Genomic DNA Kit (Tiangen) was used in all PCR experiments. The F1/R1 and F2/R2 primer sets for each gene were used to amplify an ~2.7 kb product of the targeted integration and an ~2.5 kb product of random integration. For UTX-3 × FLAG and JMJD3-3 × FLAG knock-in, we used western blotting to validate FLAG expression.

All gRNA sequences and primer sequences are listed in Supplementary Table 1.

**Neural progenitor cell (NPC) differentiation from hESCs.** Wild-type hES cells and three KDM6 mutant hES cell lines lacking UTX, JMJD3 or both at 95–100% cell confluence were seeded onto Matrigel-coated 12-well plates in mTeSR1 medium. These cells were cultured in N2B27 medium (50% DMEM/F12 (Gibco), 50% neurobasal (Gibco), 0.5 × N2 (Gibco), 0.5 × B27 (Gibco), 1% Glutamax (Gibco), 1% NEAA (Gibco), 5 μg mL$^{-1}$ insulin (Gibco), and 1 μg mL$^{-1}$ heparin (Sigma)) plus 5 μM SB431542 (Selleck) and 5 μM dorsomorphin (Selleck)[41]. The medium was changed every two days. After 8 days of induction, the cells were passaged on new six-well plates coated with Matrigel in N2B27 medium plus 10 μM Y-27632 at a 1:2 ratio. The medium was changed every 2 days. After 16 days, canonical neural rosettes appeared and were selected. Then, these neural rosettes were dissociated to single cells with Accutase (Sigma). These neural rosettes were named NPC passage 0 (NPC-P0). A total of 1 × 10$^6$ NPCs from NPC-P0 were cultured in NPC medium (N2B27 medium, 20 ng mL$^{-1}$ bFGF and 20 ng mL$^{-1}$ EGF) and subjected to a proliferation assay. Every 7 days, these NPCs were passaged and counted. The following passaged NPCs were named NPC passage 1 (NPC-P1), passage 2 (NPC-P2), passage 3 (NPC-P3), and passage 4 (NPC-P4). These NPCs were cultured in NPC medium, and the medium was changed every 2 days.

**Neural progenitor cells (NPCs) undergo random differentiation.** Wild-type NPCs and three KDM6 mutant NPCs lacking UTX, JMJD3 or both (NPC-P2) at passage two were digested into single cells and used for random differentiation[41]. NPCs (2 × 10$^4$) were seeded onto Matrigel-coated 24-well plates and cultured in NPC medium plus 10 μM Y-27632. After 1 day, these cells were cultured in N2B27 medium plus 20 ng mL$^{-1}$ GDNF (Peprotech), 20 ng mL$^{-1}$ BDNF (Peprotech), and 1 mM cAMP (Sigma). The medium was changed every 2 days. After 28 days, canonical neuronal morphology appeared, and the neurons were examined by immunofluorescence and qRT-PCR.

**Quantitative real-time PCR (qRT-PCR).** Total RNA was extracted from cells with TRIzol (Invitrogen) and reverse transcribed with a HiScript II 1st Strand cDNA Synthesis Kit (Vazyme). Then, qRT-PCR was performed with ChamQ SYBR qPCR Master Mix (Vazyme) and a CFX96 machine (Bio-Rad). *GAPDH* was used to normalize the qRT-PCR results from human samples. All data were analysed with three replicates. All primer sequences are listed in Supplementary Table 2.

**Western blot analysis.** To detect H3K27me3, H3K4me3, and PAX6, RIPA buffer (Beyotime) was used to lyse cells on ice. Whole-cell extracts were subjected to 15% SDS-PAGE. These samples were transferred to PVDF membranes (Millipore). Then, these PVDF membranes were incubated with primary antibodies at 4 °C for 12 h. After washing five times with TBST for 5 min each time, the membranes were incubated with HRP-conjugated secondary antibodies. After washing five times in TBST for 5 min each time, the membranes were detected by ECL (Beyotime) and visualized with a SmartChemi image analysis system (Sage Creation). The antibodies were used in western blot assays following the manufacturer's recommendations. Detailed information about the antibodies used is listed in Supplementary Table 3. All uncropped western blot images can be found in Supplementary Figs. 7–9.

**Flow cytometry analysis.** Accutase (Sigma) was used to digest sample cells to single cells, and then these single cells were fixed in fixation buffer (BD Biosciences) at room temperature for ~20 min. After fixation, the cells were washed in PBS and

then permeabilized in perm/wash buffer (BD Biosciences) for 10–15 min at 4 °C. After washing with PBS, the cells were incubated with the corresponding primary antibodies at 37 °C for 30 min. Meanwhile, other cell samples were incubated with corresponding isotype control antibodies at 37 °C for 30 min. After incubation, the cells were washed with PBS. Then, these samples were incubated with secondary antibodies at 37 °C for 30 min. After washing twice, the cells were resuspended in PBS and analysed with an Accuri C6 flow cytometer (BD Biosciences). Detailed information about the antibodies used is listed in Supplementary Table 3. A graphical account of all FACS sequential gating/sorting strategies is provided in Supplementary Figs. 10–14.

**Immuno-staining assay.** Cells for the immuno-staining assay were plated onto Matrigel-coated 24-well plates. The cells were fixed in 4% paraformaldehyde (PFA) at room temperature for ~20 min. After washing three times with PBS, the cells were permeabilized with 0.3% Triton X-100 (Sigma) and 10% goat serum in PBS and incubated with primary antibodies for 12 h at 4 °C. After washing three times, the cells were incubated with secondary antibodies at room temperature for 1 h. After washing three additional times, the cells were stained with DAPI (Sigma) for 5 min at room temperature. Images of the immunostained samples were captured with an LSM 800 microscope (Zeiss). Detailed information about the antibodies used is listed in Supplementary Table 3.

**EdU assay and cell cycle and apoptosis analyses.** An EdU assay was performed with a Click-iT™ EdU Pacific Blue™ flow cytometry assay kit (Invitrogen) according to the manufacturer's recommendations. A total of 1 × 10$^6$ wild-type NPCs and three KDM6 mutant NPC lines lacking UTX, JMJD3 or both were plated onto 6-well plates in NPC medium with 10 μM EdU or without EdU as a negative control for ~16 h. These cells were digested to obtain single cells, and the cells were fixed in fixation buffer at room temperature for 20 min. Then, the cells were washed in PBS and permeabilized in perm/wash buffer for 10–15 min at 4 °C. After washing, the cells were incubated in PBS with CuSO$_4$, the fluorescent dye picolyl azide, and reaction buffer additive for 1 h at room temperature. Then, the samples were analysed with an Accuri C6 flow cytometer.

Cell cycle assays were performed with a cell cycle detection kit (Keygen) according to the manufacturer's recommendations. Wild-type and three KDM6 mutant NPC lines were digested to obtain single cells, and 1 × 10$^6$ cells were fixed in fixation buffer at room temperature for ~20 min. After washing, the cells were incubated with RNase at 37 °C for 30 min, following which propidium iodide (PI) was added, and the cells were incubated at 4 °C for 30 min. Then, these samples were analysed with an Accuri C6 flow cytometer.

Apoptosis analysis was performed with an Annexin V-FITC/PI cell apoptosis detection kit (Keygen) according to the manufacturer's recommendations. Wild-type NPCs and three KDM6 mutant NPC lines were digested to obtain single cells, and 1 × 10$^5$ cells were incubated with binding buffer, Annexin V-FITC, and PI at room temperature for 5–15 min. Then, the samples were analysed with an Accuri C6 flow cytometer.

**RNA-seq and rank correlation and heatmap analyses.** Wild-type and KDM6-dKO NPCs and their randomly differentiated cells at day 28 (D28) were lysed with TRIzol (Invitrogen). Total RNA was extracted according to the manufacturer's recommendations. Then, sequencing libraries were established with a TruSeq RNA sample preparation kit (Illumina) according to the manufacturer's recommendations. The samples were run on a NextSeq system with a NextSeq 500 Mid Output kit (Illumina).

Then, all RNA-Seq data were analysed. Briefly, reads were aligned to the human genome (UCSC hg38) using HISAT2 (v2.0.4), and gene expression was determined using SAMtools (v1.3.1) and htseq-count (v0.6.0), filtered by a threshold of at least 20 average raw read counts among samples, and then normalized by GC content and gene length using EDASeq (v2.12.0). Differential expression was determined, and PCA plots were prepared using DESeq (v 1.18.1); differences in gene expression with a *p* value < 0.05 and a fold-change > 2 were considered significant differences. A correlation plot was prepared using ggplot2 (v2.2.1), and a heatmap was prepared using pheatmap (v1.0.10). Gene ontology analysis was performed using clusterProfiler (v3.6.0).

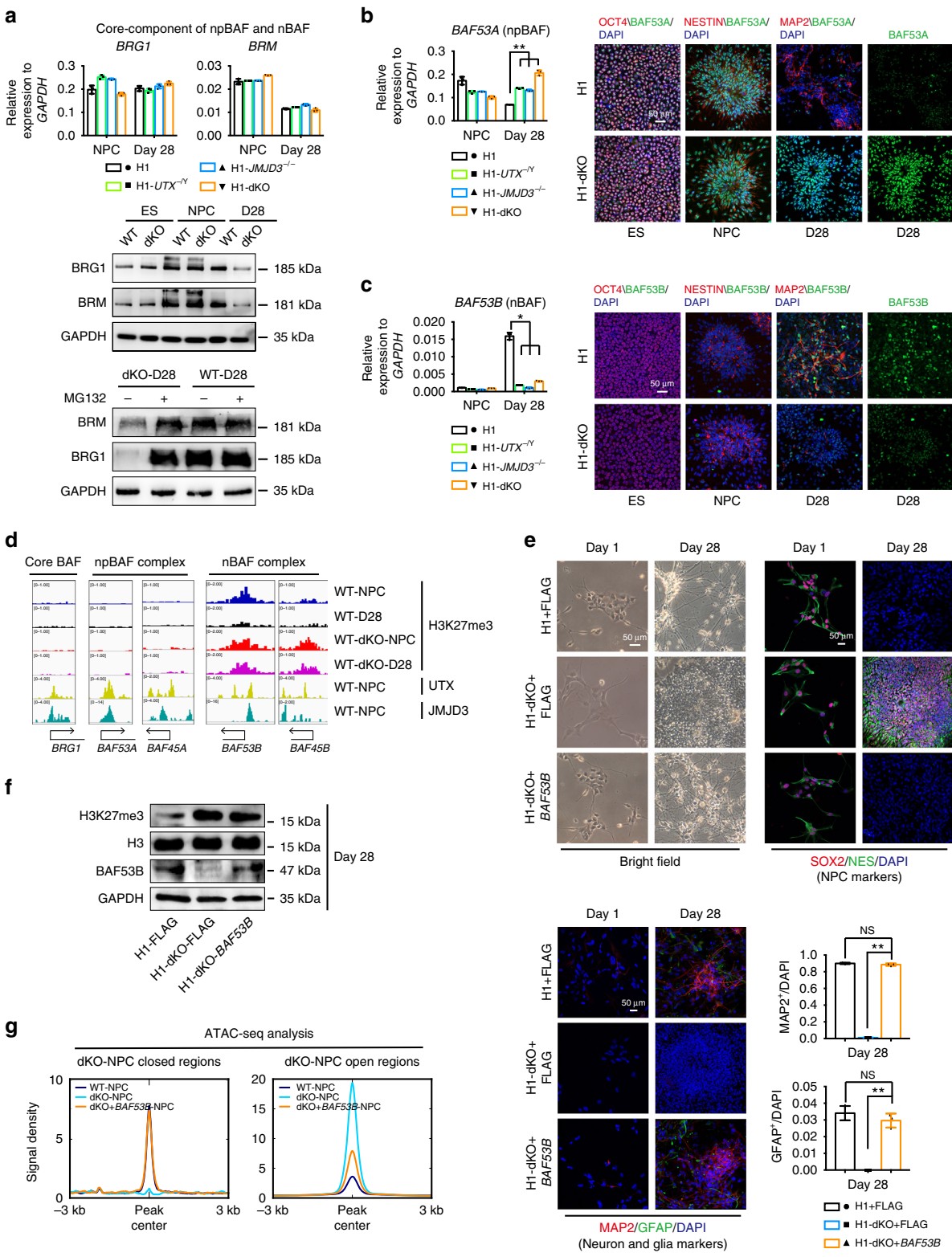

**ATAC-seq**. ATAC-seq and data analysis were performed. In brief, $5 \times 10^4$ cells from each sample were processed and then used to generate DNA libraries with a Nextera DNA library preparation kit (Illumina) according to the manufacturer's recommendations. These DNA libraries were used for sequencing on a NextSeq 500 platform. Adaptors were cut from reads using cutadapt (v1.13), and reads were aligned to the human genome (UCSC hg38) using Bowtie2 (v2.2.5). Duplicates were removed using SAMtools (v1.3.1) and Picard tools (v1.90). Signals were compiled using MACS2 callpeak and bdgcmp, and differential open peaks were called using MACS2 bdgdiff[62]. A signal density heatmap and profile were plotted using deepTools (v2.4.2), and motifs were found using homer. Peaks were annotated with gene annotation using ChIPpeakAnno (v3.12.7), and gene ontology analysis was performed using clusterProfiler (v3.6.0).

**ChIP-seq**. ChIP experiments and data analysis were performed. In brief, $1 \times 10^7$ cells from each sample were crosslinked in 1% formaldehyde with rotation for 10 min at room temperature. Then, these crosslinking reactions were stopped with incubation with 0.125 M glycine with rotation for 5 min at room temperature. After

**Fig. 7 BAF53B rescues the differentiation defect in JMJD3/UTX-deficient human NPCs. a** Upper panel, qRT-PCR and western blot analysis of BRG1 and BRM in the indicated ESCs, NPCs, and their differentiated cells at day 28. Bottom panel, western blot analysis of BRG1 and BRM in the indicated differentiated wild-type (WT) or dKO cells at day 28 with the proteasome inhibitor MG132. The data represent the mean ± SD (standard deviation) from three independent replicates ($n = 3$). npBAF, neural progenitor cell-specific BAF complex. nBAF, neuron-specific BAF complex. dKO, deletion of both JMJD3 and UTX. **b** qRT-PCR and immuno-staining analysis of the npBAF component BAF53A in the indicated ESCs, NPCs, and their differentiated cells at day 28. The significance level was determined using unpaired two-tailed Student's t-tests. **$P < 0.01$. The data represent the mean ± SD from three independent replicates ($n = 3$). **c** qRT-PCR and immuno-staining analysis of the nBAF component BAF53B in the indicated ESCs, NPCs, and their differentiated cells at day 28. The significance level was determined using unpaired two-tailed Student's t-tests. *, $P < 0.05$. The data represent the mean ± SD from three independent replicates ($n = 3$). **d** Genomic views of the H3K27me3 modification and UTX and JMJD3 binding of BAF genes in the indicated cell lines. **e** Morphological examination and immuno-staining for the NPC markers SOX2/NES, the neuronal/glial markers MAP2/GFAP in dKO NPCs (H1-dKO + BAF53B) with forced BAF53B expression during random differentiation at day 1 and day 28. The numbers of MAP2$^+$ or GFAP$^+$ cells were analysed. Significance was determined by unpaired two-tailed Student's t-tests. **$P < 0.01$. The data represent the mean ± SD from three independent replicates ($n = 3$). Scale bar, 50 μm. **f** Western blotting for H3K27me3 or BAF53B in the indicated wild-type, dKO, or BAF53B-rescued dKO differentiated cells at day 28. **g** Signal densities for OC or CO regions in the indicated wild-type (WT), dKO, or BAF53B-rescued dKO differentiated cells at day 28 from ATAC-seq data. OC, open-to-closed, indicates regions in dKO NPCs with reduced chromatin accessibility. CO, closed-to-open, indicates regions in dKO NPCs with increased chromatin accessibility. Error bars throughout the figure represent the SD (standard deviation) from three independent replicates ($n = 3$). Source data are provided as a Source Data file. More information is also provided in Supplementary Fig. 5. NS no significance.

washing twice with cooled PBS, the cells were sonicated in 1% SDS lysis buffer plus 1 mM PMSF and protease inhibitor cocktail to obtain 200–500 bp chromatin fragments, and these sonicated samples were dialyzed with ChIP dilution buffer. The sonicated fragments were incubated with magnetic beads (Dynabeads protein A and G (1:1)) (Invitrogen) and 5 μg of anti-H3K27me3 antibody (Millipore, 17–622) or anti-FLAG antibody (Sigma-Aldrich, M8823) with rotation overnight at 4 °C. These antibody-bound complexes were washed with rotation for 5 min with the following buffers in sequence: low-salt wash buffer (0.1% SDS, 1% Triton X-100, 2 mM EDTA, 20 mM Tris-HCl (pH 8.0), 150 mM NaCl), high-salt wash buffer (0.1% SDS, 1% Triton X-100, 2 mM EDTA, 20 mM Tris-HCl (pH 8.0), 500 mM NaCl), LiCl wash buffer (0.25 M LiCl, 1% IGEPAL-CA630, 1% deoxycholic acid (sodium salt), 1 mM EDTA, 10 mM Tris-HCl (pH 8.0)), and TE buffer (10 mM Tris-HCl (pH 8.0), 1 mM EDTA). These complexes were reverse crosslinked, and the ChIPed DNA samples were purified for ChIP-Seq. Approximately 10 ng of ChIPed DNA and corresponding input DNA, as measured by a Qubit fluorometer (Invitrogen), were used to generate DNA libraries using a ChIP-seq sample preparation kit (Illumina), and these DNA libraries were then sequenced on a NextSeq 500 platform.

For H3K27me3 ChIP analysis, sequenced reads were base-called and de-multiplexed using standard Illumina software. Bowtie2[63] was used to align the reads to the human genome (hg38). Reads mapped to identical positions with the same orientation in the genome were collapsed into one read. Regions of H3K27me3 enrichment (peaks) were called using the SICER software package[64], with the input genomic DNA used as a background control (parameters: W = 200; G = 600; FDR cut-off of 0.01 applied). Binding profiles and heatmaps were generated using ngsplot[65] and deepTools[66], respectively. Gene ontology analysis was performed using the DAVID functional annotation tool.

For UTX-FLAG and JMJD3-FLAG ChIP analysis, the sequencing reads were filtered by Trimmomatic (v0.35) and then mapped to the human reference sequence for UCSC (hg38) using Bowtie2 (v2.2.5). Uniquely mapped reads were retained using SAMtools (v1.3.1) (with the parameters -F 1804 -f 2 -q 30) and Picard tools MarkDuplicates (1.90). We performed peak calling of transcription factor ChIP-seq data using the MACS2 (v2.1.0) callpeak module (with the parameters -p 0.1 --nomodel --extsize 150 -B --SPMR --keep-dup all --call-summits), and then only peaks with a $q$ value < 0.05 were retained. Signal tracks were computed using the MACS2 bdgcmp module with the parameter -m ppois). The bigWig signal files were visualized using the computeMatrix, plotHeatmap and plotProfile modules in DeepTools (v2.4.2). ChIPpeakAnno (v3.16.1) was used to identify nearby genes from the peaks obtained from MACS. Gene Ontology (GO) analysis was performed using clusterProfiler (v3.10.1).

**Statistical analysis**. In general, the results are presented as the mean ± SD (standard deviation) calculated using Microsoft Excel and GraphPad Prism from at least three biological repeats. The significance level between samples was determined using unpaired two-tailed Student's t-tests. Differences with a $P$ value < 0.05 were considered statistically significant, as indicated in the figures. No samples were excluded for any analysis.

**Reporting summary**. Further information on research design is available in the Nature Research Reporting Summary linked to this article.

## Data availability
The RNA-Seq, ATAC-seq, and ChIP-seq data have been deposited in the Gene Expression Omnibus database under the accession codes GSE118999 and GSE133209. A

reporting summary for this article is available as a Supplementary Information file. PCR and qRT-PCR data have also been deposited in Figshare (https://doi.org/10.6084/m9.figshare.8035574). The source data underlying Figs. 1c, 2d, f, 3e and 7a–c and Supplementary Figs. 1e, f, 2c, 6c and e are provided as a Source Data file. The authors declare that all data supporting the findings of this study are available within the article and its Supplementary information files or from the corresponding author upon reasonable request.

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

## Acknowledgements

We thank the lab members in GIBH for their kindly help. This work was supported by the National Key Research and Development Program of China, Stem Cell and Translational Research (2017YFA0102600); the National Natural Science Foundation of China (31801220, 31421004, 31500791, 81571238, 31801225, 81672261, 81700149); China Postdoctoral Science Foundation Funded Project (2019M652953); Strategic Priority Research Program of Chinese Academy of Sciences (XDA16030201); Natural Science Foundation of Guangdong Province, China (2016A030313167); Cooperation Grant of Natural Science Foundation of Guangdong Province (2014A030312012, 2017A030310376); the Science and Information Technology of Guangzhou Key Project (201904020045); Science and Technology Program of Guangzhou (201904010462, 201804010339, 201508020257); Hundred-talented program from Chinese Academy of Sciences (to X.Z.); the Guangdong Province Special Program for Elite Scientists in Science and Technology Innovation (to G.P., 2015TX01R203); the Informationization Special Project of Chinese Academy of Sciences, E-Science Application for Knowledge Discovery in Stem Cells (XXH13506-203); Innovative Team Program of Guangzhou Regenerative Medicine and Health Guangdong Laboratory (2018GZR110104005); the Open Research Funds of the State Key Laboratory of Ophthalmology (2019KF06).

## Author contributions

G.P. and Y.S. initiated and designed the project, and wrote the manuscript. Y.S., Y.Z. and Y.Z. performed most experiments and analyzed result data. Y.Z. performed the ChIP sequencing. J.Z. performed KDM6 rescue experiment. Z.L., J.Y., N.M. and C.Z. performed the gene knockout of the KDM6 family in hESCs. W.H. performed NPC differentiation from hESCs. Q.C. performed the RNA sequencing. Z.S., C.L. and S.S. performed the FACS and RT-qPCR. K.H., T.Z., Y.Z. and C.W. performed the ATAC sequencing. T.Z. and W.S. performed karyotype analysis. T.W. and J.N. analyzed the RNA-seq, ChIP-seq and ATAC-seq data. M.Z., X.Z., B.L., Y.W. and D.P. gave suggestions about experiments and the manuscript. All authors read and approved the final manuscript.

## Competing interests

The authors declare no competing interests.
