## [Peer Review File · Nature Communications]

Reviewers' comments:

Reviewer #1 (Remarks to the Author):

In this study entitled "KDM6s Determine Fidelity and Lineage Specification of Human Neural Progenitor Cells" the authors determined the role of the histone demethylases UTX and JMJD3 in human neurogenesis. They generated UTX, JMJD3 and double knock-out UTX/ JMJD3 (dKO) hESCs to interrogate their function in neuronal differentiation. They found that while both UTX and JMJD3 are dispensable for hESC self-renewal and differentiation towards neuronal precursor cells (NPC), dKO-NPC cells cannot be propagated and fail to terminally differentiate. They found that chromatin accessibility and expression of neuronal genes is severely compromised in dKO cells. Interestingly, transition from NPCs to differentiated cells requires a massive chromatin compaction, which is diminished in dKO cells. H3K27me3 ChIP-seq in dKO cells revealed an aberrant accumulation of this repressive mark in ~1000 genes, associated to neurogenesis. Several BAF complex subunits are silenced by H3K27me3 in dKO cells, suggesting that absence of BAF complexes in dKO cells may be responsible for the lack differentiation. Indeed, the authors show that ectopic expression of the neural BAF complex subunit BAF53B rescues the differentiation defects of the dKO cells. This is an excellent manuscript. Overall, the results are clear and well presented, and the conclusions are well supported by the data.

The manuscript has numerous problems of grammar and usage that should be corrected. That said, I only have the following minor points that need to be clarified before I can recommend publication of the paper:

1. Supplemental Figures 1 and 5 are barely discussed in the text. Please describe the results accordingly. In Supplemental Figure 4, the enriched signal of H3K27me3 peaks in WT-D28 is not convincing.
2. It would be nice to know the correlation between gene expression and chromatin accessibility at the deregulated genes in dKO-NPC cells. Also, the authors should discuss the potential role of AP1/JUN in dKO -NPCs.
3. The observation that the chromatin is mainly closing from the transition of NPCs to D28 is very interesting, and supports a role of H3K27me3 during terminal differentiation. ~14000 genes are closed while ~1600 are repressed (Figure 5). The correlation of chromatin accessibility changes with gene expression changes should be also included. Indeed, these results suggest that chromatin is closing without affecting gene expression. Are the ~1600 genes repressed decorated with H3K27me3 and also compacted? Moreover, only 10 genes become open in WT-D28 cells (Figure 5e) but 931 genes have less H3K27me3 signal (Figure 6d). These results indicate that chromatin compaction and H3K27me3 levels are poorly correlated. Please discuss.
4. Western blots of Flag-UTX and Flag-JMJD3, and Flag-BAF53B should be included.

Reviewer #2 (Remarks to the Author):

In the study, the authors tried to show that the H3K27me3 demethylases, UTX and JMJD3, are essential for fate commitment of neural stem or progenitor cells. They used in vitro model system and many known methods to show that loss of UTX and JMJD3 perturbs neuronal and astrocyte generation (differentiation) from hESC-derived neural progenitor cells (NPCs). They also assert UTX and JMJD3 act ahead of other epigenetic factors like the BAF complex and signaling pathways known to regulate neurogenesis. As a result; forced expression of BAF53B, one of several neuron-specific BAF complex subunits, rescued the aberrant neuron and glia fate commitment of UTX and JMJD3-deficient neural

progenitor cells. While the study is generally interesting, there are major gaps and over-interpretation of the results in the study that warrants major revision.

Major comments:

1. Can authors explain why despite alterations in chromatin accessibility in dKOs, mRNA levels of BAF complex subunits remained unchanged but protein levels were reduced in D28?
2. In possible connection with comment 2, could authors be less emphatic on the implication that UTX and JMJD3 regulate the expression of BAF complex subunits, especially when there is no obvious result to support it or rather a confounding one to show for it?
3. How does the known interaction of UTX and JMJD3 with the scaffolding BAF subunits 155 and 170 (Miller et al., 2010, Mol Cell; Narayanan et al., 2015, Cell Reports; Nguyen et al., 2018, Stem Cell Reports) influence the interpretation of the outcome of this study?
4. Authors should examine possibility whether BAF53B in the rescue experiment might influence on H3K27me3 demethylase activity of KDM6A/B
5. Does npBAF subunits (e.g. BAF53A) rescue proliferation defects?
6. Authors should have considered using an in vivo model system to corroborate their findings. It would be interesting if the phenotype achieved in the dKO is reproducible in
7. Authors should examine apoptosis not only in NPC, but also in neurons, glia cells.
8. In Figure 2. KDM6s are essential for proliferation, authors should present few cell cycle genes, which are direct targets of KDM6A/B, H3K27me3.
9. vivo or at least ex vivo in say rodent cortex. This should be done, unless there exists human and rodent interspecies difference in H3K27me3 demethylase function.

Minor comments

1. The micrographs in Figure 1f does not well represent the poor proliferation due to loss of UTX and JMJD3. The Pax6/Ki67 immunostaining for instance is misleading. Authors should provide images that more convincingly reflect the said proliferation disturbance in the KOs.
2. It would be informative if you could explain why in spite of ablation of H3K27me3 demethylases in hESC, escBAF transitioned 'normally' to npBAF to give rise to NPC but the latter could not switch to the nBAF? Could there be a possible chromatin remodeling-independent function of UTX and JMJD3 that you are missing (e.g. Miller et al., 2010, Mol Cell)?
3. Fig. 2C: why Utx_KO has more profound effect than dKO

Reviewer #3 (Remarks to the Author):

In their manuscript, Guangjin Pan and colleagues have studied the function of KDM6A (UTX) and KDM6B (JMJD3) in human embryonic stem cells (hESCs) as they differentiate into neural cell types in vitro. The authors generated UTX and JMJD3 knockouts (KOs) in hESCs and studied the consequences of UTX-KO, JMJD3-KO, and double UTX/JMJD3-KO (dKO). Without KDMs, hESCs could undergo neural induction normally, but these neural precursor cells (NPCs) were defective for long-term proliferation, exhibiting increased levels of apoptosis. After withdrawal of growth factors, normal NPCs can generate neurons and glia, but such neurogenesis/gliogenesis was defective in cells lacking UTX, JMJD3 or both. Chromatin analysis with ChIP-seq and ATAC-seq indicated that loss of KDMs leads to increased levels of H3K27me3 (a modification associated with gene expression) as well as decreased accessibility (as determined by ATAC-seq) on genes that correlate with neurogenesis. Overexpression of a BAF complex component – BAF53B – rescued neuronal/glial differentiation in KDM6-deficient NPCs.

While the function of chromatin-modifying factors is of significant interest to the field of neural development, the major conclusions in this manuscript do not significantly advance our overall

understanding of UTX and JMJD3 function. For instance, knockout of UTX has already been shown to affect neural stem cell proliferation and differentiation in the developing mouse cortex (Lei and Jiao, *Stem Cell Reports*, 2018). Knockout of JMJD3 in neural stem cells *in vivo* also impairs neurogenesis (Park et al., *Cell Reports*, 2014). In mouse ESCs, knockdown of JMJD3 impairs neural cell commitment (Burgold et al., *PLOS ONE*, 2009), which is a result that is somewhat different from the findings in this manuscript but not discussed by the authors. Thus, the overall concept that UTX and JMJD3 are important for neurogenesis and lineage specification has already been clearly demonstrated in mice. Coming to a similar set of conclusions in human NPCs *in vitro* seems fairly incremental.

The genome-wide analyses (H3K27me3 ChIP-seq, RNA-seq, and ATAC-seq) of JMJD3 and UTX knockouts should be complemented by JMJD3 and UTX ChIP-seq. Without knowledge of where JMJD3 and UTX are localized on the genome, it is difficult to interpret the chromatin and transcriptomic data presented in this manuscript. That is, simply observing increased levels of H3K27me3 and decreased accessibility at a particular locus does not demonstrate that JMJD3 and/or UTX normally functions at that locus. For instance, it is possible that EZH2 (the H3K27me3 methyltransferase) activity at those genes increases with KDM6-deficiency. Of note, JMJD3 ChIP-seq has been achieved in studies of mouse NSCs (e.g., Fueyo et al., *Nuc Acid Res*, 2018; Estaras et al., *MCB* 2013), which has helped our understanding of JMJD3 mechanism(s) in transcriptional activation. For instance, in addition to demethylation of H3K27me3 at promoters, KDM6s have important function at enhancers as well as in transcriptional elongation. These additional mechanisms should be considered.

Other comments:

1. Does knockdown of BAF53a in dKO cells rescue neurogenesis?
2. When it is overexpressed, does BAF53b localize to expected targets? Or, does BAF53b simply put the entire cell into a generally more transcriptionally permissive state?

Reviewers' comments:

Reviewer #1 (Remarks to the Author):

In this study entitled “KDM6s Determine Fidelity and Lineage Specification of Human Neural Progenitor Cells” the authors determined the role of the histone demethylases UTX and JMJD3 in human neurogenesis. They generated UTX, JMJD3 and double knock-out UTX/ JMJD3 (dKO) hESCs to interrogate their function in neuronal differentiation. They found that while both UTX and JMJD3 are dispensable for hESC self-renewal and differentiation towards neuronal precursor cells (NPC), dKO-NPC cells cannot be propagated and fail to terminally differentiate. They found that chromatin accessibility and expression of neuronal genes is severely compromised in dKO cells. Interestingly, transition from NPCs to differentiated cells requires a massive chromatin compaction, which is diminished in dKO cells. H3K27me3 ChIP-seq in dKO cells revealed an aberrant accumulation of this repressive mark in ~1000 genes, associated to neurogenesis. Several BAF complex subunits are silenced by H3K27me3 in dKO cells, suggesting that absence of BAF complexes in dKO cells may be responsible for the lack of differentiation. Indeed, the authors show that ectopic expression of the neural BAF complex subunit BAF53B rescues the differentiation defects of the dKO cells. This is an excellent manuscript. Overall, the results are clear and well presented, and the conclusions are well supported by the data.

RE: We thank the reviewer for the positive comments.

The manuscript has numerous problems of grammar and usage that should be corrected. That said, I only have the following minor points that need to be clarified before I can recommend publication of the paper:

RE: Thanks for this suggestion. We revised the text.

1. Supplemental Figures 1 and 5 are barely discussed in the text. Please describe the results accordingly. In Supplemental Figure 4, the enriched signal of H3K27me3 peaks in WT-D28 is not convincing.

RE: Thanks for this suggestion. We revised text and described supplementary Figure 1 and 5 (arranged as Supplementary 6) in the main text. Yes, there were not many genes that gain more H3K27me3 in WT-D28, so the overall intensity of these genes looks weaker compared with other gene lists (Supplemental Figure 4a). However, the selected individual gene did show clear enrichment on H3K27me3 in WT-D28 (Supplementary Figure 4b).

2. It would be nice to know the correlation between gene expression and chromatin accessibility at the deregulated genes in dKO-NPC cells. Also, the authors should discuss the potential role of AP1/JUN in dKO -NPCs.

RE: Thanks for this suggestion. As suggested, we analyzed the correlation between gene expression and chromatin accessibility on the changed genes in dKO-NPC cells. The

downregulated genes in dKO NPCs showed obvious reduced chromatin accessibility and increased H3K27me3 enrichment in dKO NPCs (Figure 4f upper panel, Supplementary Figure 4d left panel) , suggesting a co-relationship between chromatin accessibility and H3K27me3 modification. Consistently, the upregulated genes in dKO NPCs showed similar chromatin accessibility and H3K27me3 modification between WT and dKO cells (Figure 4f bottom panel, Supplementary Figure 4d right panel).

Yes, the AP1/JUN family proteins, including FOS, ATF, JUN, MAF, and CREB family members, are mainly involved in cell transformation, proliferation, differentiation, survival and apoptosis (M. Ameyar, et al, Biochimie, 2003, P. Madrigal and K. Alasoo, Trends in Cell Biology, 2018). Our results showed that chromatin regions enriched in AP1 motifs are much more open in dKO-NPCs than the WT cells, which might in part explain the proliferation defect and apoptosis in dKO NPCs. We discussed these potential role of AP1/JUN in dKO-NPCs in revised text. Thanks for this suggestion.

3. The observation that the chromatin is mainly closing from the transition of NPCs to D28 is very interesting, and supports a role of H3K27me3 during terminal differentiation. ~14000 genes are closed while ~1600 are repressed (Figure 5). The correlation of chromatin accessibility changes with gene expression changes should be also included. Indeed, these results suggest that chromatin is closing without affecting gene expression. Are the ~1600 genes repressed decorated with H3K27me3 and also compacted? Moreover, only 10 genes become open in WT-D28 cells (Figure 5e) but 931 genes have less H3K27me3 signal (Figure 6d). These results indicate that chromatin compaction and H3K27me3 levels are poorly correlated. Please discuss.

RE: Thanks for this suggestion. These are good points and we re-analyzed the data. The 1651 gene list that were downregulated in WT-D28 differentiated cells indeed exhibit significantly reduced chromatin accessibility and increased H3K27me3 compared with undifferentiated NPCs (Figure 5f, Supplementary Figure 4e). These data indicate that even though many genes showed reduced chromatin accessibility but were not transcriptionally silenced at a given time point, those genes with reduced expression were certainly closed on their chromatin. Consistently, those genes losing H3K27me3 in differentiated neural cells did show significantly upregulated expression in WT D28 differentiated neural cells (Supplementary Figure 4c, left panel). We had discussed these new data in revised text.

4. Western blots of Flag-UTX and Flag-JMJD3, and Flag-BAF53B should be included.

RE: Thanks for this suggestion. The western blots of Flag-UTX and Flag-JMJD3, and Flag-BAF53B, were provided in Supplemental Figure 3b and Figure 7f, respectively.

Reviewer #2 (Remarks to the Author):

In the study, the authors tried to show that the H3K27me3 demethylases, UTX and JMJD3, are essential for fate commitment of neural stem or progenitor cells. They used in vitro model system and many known methods to show that loss of UTX and JMJD3 perturbs neuronal and astrocyte generation (differentiation) from hESC-derived neural progenitor cells (NPCs). They also assert UTX and JMJD3 act ahead of other epigenetic factors like the BAF complex and signaling pathways known to regulate neurogenesis. As a result; forced expression of BAF53B, one of several neuron-specific BAF complex subunits, rescued the aberrant neuron and glia fate commitment of UTX and JMJD3-deficient neural progenitor cells. While the study is generally interesting, there are major gaps and over-interpretation of the results in the study that warrants major revision.

RE: We thank the reviewer for the positive comment.

Major comments:

1. Can authors explain why despite alterations in chromatin accessibility in dKOs, mRNA levels of BAF complex subunits remained unchanged but protein levels were reduced in D28?

RE: Thanks for this suggestion. Yes, our data did show that mRNA levels of BRG1 and BRM, were less changed but the protein levels were significantly reduced in D28 differentiated neural cells (Figure 7a). These data indicate that these BAF units might undergo degradation in D28 differentiated cells in the absence of KDM6s. Indeed, treatment with MG132, a proteasome inhibitor obviously restored the protein level in dKO-D28 differentiated neural cells. Furthermore, we also performed more experiments and showed KDM6, either JMJD3 or UTX could maintain the protein stability of these core BAF core units (Supplementary Figure 5a). These new analyses and data provide another molecular mechanism on the role of KDM6s in neurogenesis. KDM6s maintain the stability of BAF complex during neurogenesis to ensure fate transition from NPCs to subtype neural cells. Thanks again for the useful comments.

2. In possible connection with comment 2, could authors be less emphatic on the implication that UTX and JMJD3 regulate the expression of BAF complex subunits, especially when there is no obvious result to support it or rather a confounding one to show for it?

RE: Thanks for this suggestion. Based on the new analyses, the subunits in BAF complex were differentially regulated by KDM6s during neurogenesis. The expression of the core units such as BRG1, BRM and NPC specific BAFs such as BAF53A, do not require KDM6 function (Figure 7a and 7b). Accordingly, no H3K27me3 enrichment or accumulation were found on these genes in either WT or dKO cells (Figure 7d). However, the neuron specific BAF units, such as BAF53B, definitely need KDM6s for their proper expression during subtype neural differentiation (Figure 7c). Consistently, these nBAF genes showed significant H3K27me3 enrichment that need KDM6 to remove during differentiation (Figure 7d). The differential requirement of KDM6 in expression of npBAF and nBAF might explain KDM6s are essential in subtype neural cell specifications rather than NPC generation. We revised the manuscript and discuss these mechanisms on the role of KDM6s in neurogenesis based on human model.

3. How does the known interaction of UTX and JMJD3 with the scaffolding BAF subunits 155 and 170 (Miller et al., 2010, Mol Cell; Narayanan et al., 2015, Cell Reports; Nguyen et al., 2018, Stem Cell Reports) influence the interpretation of the outcome of this study?

RE: Thanks for this suggestion. These previous reports suggest that BAF subunits 155 and 170 interact with KDM6s and potentiate their H3K27me3 demethylase activity. In our results, we found that KDM6s conversely maintain protein stability of core BAF complex subunits, BRG1 and BRM (Figure 7a and Supplementary Figure 5a). Also, we found KDM6s are required for the expression of nBAF that is essential to neuron specification (Figure 7b-7g). These publications together with our new data strongly suggest that the neural BAFs and KDM6s are largely inter-dependent to ensure the fate transition during neurogenesis. Thanks for this comment and discussed this in the revised text.

4. Authors should examine possibility whether BAF53B in the rescue experiment might influence on H3K27me3 demethylase activity of KDM6A/B

RE: Thanks for this suggestion. Yes, this is a good point. We examined H3K27me3 level in BAF53B rescued cells. However, H3K27me3 still showed obvious accumulation in BAF53B rescued dKO cells (Figure 7f), indicating BAF53B drive neuron fate transition independent of H3K27me3. Furthermore, we also showed that BAF53B largely rescued “mis-open or closed” chromatin accessibility by dKO of KDM6s (Figure 7g), thus enable the neuron fate specification of dKO NPCs. These data provide an explanation on how nBAF drive neuron fate transition independent of KDM6s. We have included this new data in the revised version.

5. Does npBAF subunits (e.g. BAF53A) rescue proliferation defects?

RE: Thanks for this suggestion. We performed experiment and overexpressed npBAF subunit BAF53A in dKO NPCs. However, these NPCs also exhibited severe proliferation defect as dKO NPCs at higher passages (Supplementary Figure 5b and 5c). We discussed this in the revised version.

6. Authors should have considered using an *in vivo* model system to corroborate their findings. It would be interesting if the phenotype achieved in the dKO is reproducible in

RE: Thanks for this suggestion. The current focus is to understand the role of KDM6s during fate transition from human pluripotent stem cells to NPCs and then subtype neurons and glia cells. Examining the role of KDM6s using an *in vivo* model would be interesting and worth to pursue in future project.

7. Authors should examine apoptosis not only in NPC, but also in neurons, glia cells.,

RE: Thanks for this suggestion. We examined apoptosis in neurons and glia cells by ANNEXIN V immunostaining (Supplementary Figure 3a). We failed to detect significantly apoptosis in differentiation of dKO-NPCs and WT-NPCs.

8. In Figure 2. KDM6s are essential for proliferation, authors should present few cell cycle genes, which are direct targets of KDM6A/B, H3K27me3.

RE: Thanks for this suggestion. We included RNA-seq data of cell cycle genes in Figure 2j. These genes are indeed downregulated in dKO NPCs at higher passages.

9. *in vivo* or at least *ex vivo* in say rodent cortex. This should be done, unless there exists human and rodent interspecies difference in H3K27me3 demethylase function.

RE: Thanks for this suggestion. Response as above to comment “6”.

Minor comments

1. The micrographs in Figure 1f does not well represent the poor proliferation due to loss of UTX and JMJD3. The Pax6/Ki67 immunostaining for instance is misleading. Authors should provide images that more convincingly reflect the said proliferation disturbance in the KOs.

RE: Thanks for this suggestion. Our data showed that KDM6 dKO NPCs exhibited proliferation defect during long term culture *in vitro* (Figure 2). Figure 1f showed rosette like NPCs just differentiated from the pluripotent stem cells and these cells can undergo limited proliferation, that’s why stained by Ki67. We agree this is a little bit misleading and revised the text to make it more clear.

2. It would be informative if you could explain why in spite of ablation of H3K27me3 demethylases in hESC, escBAF transitioned ‘normally’ to npBAF to give rise to NPC but the latter could not switch to the nBAF? Could there be a possible chromatin remodeling-independent function of UTX and JMJD3 that you are missing (e.g. Miller et al., 2010, Mol Cell)?

RE: Thanks for this suggestion. This is an important question. As we responded to “comment 2” above, our new data showed that the subunits of BAFs are differentially regulated by KDM6s (Figure 7b-7d). While the npBAFs do not require KDM6 for their expression, the nBAFs definitely need KDM6s for their proper expression, which might explain the differential requirement of KDM6s in NPCs and neuron fate transition.

3. Fig. 2C: why Utx_KO has more profound effect than dKO

RE: Thanks for this suggestion. We think this might be due to the batch to batch variations.

Reviewer #3 (Remarks to the Author):

In their manuscript, Guangjin Pan and colleagues have studied the function of KDM6A (UTX) and KDM6B (JMJD3) in human embryonic stem cells (hESCs) as they differentiate into neural cell types in vitro. The authors generated UTX and JMJD3 knockouts (KOs) in hESCs and studied the consequences of UTX-KO, JMJD3-KO, and double UTX/JMJD3-KO (dKO). Without KDMs, hESCs could undergo neural induction normally, but these neural precursor cells (NPCs) were defective for long-term proliferation, exhibiting increased levels of apoptosis. After withdrawal of growth factors, normal NPCs can generate neurons and glia, but such neurogenesis/gliogenesis was defective in cells lacking UTX, JMJD3 or both. Chromatin analysis with ChIP-seq and ATAC-seq indicated that loss of KDMs leads to increased levels of H3K27me3 (a modification associated with gene expression) as well as decreased accessibility (as determined by ATAC-seq) on genes that correlate with neurogenesis. Overexpression of a BAF complex component – BAF53B – rescued neuronal/glial differentiation in KDM6-deficient NPCs.

RE: We thank the reviewer for the positive comment.

While the function of chromatin-modifying factors is of significant interest to the field of neural development, the major conclusions in this manuscript do not significantly advance our overall understanding of UTX and JMJD3 function. For instance, knockout of UTX has already been shown to affect neural stem cell proliferation and differentiation in the developing mouse cortex (Lei and Jiao, Stem Cell Reports, 2018). Knockout of JMJD3 in neural stem cells in vivo also impairs neurogenesis (Park et al., Cell Reports, 2014). In mouse ESCs, knockdown of JMJD3 impairs neural cell commitment (Burgold et al., PLOS ONE, 2009), which is a result that is somewhat different from the findings in this manuscript but not discussed by the authors. Thus, the overall concept that UTX and JMJD3 are important for neurogenesis and lineage specification has already been clearly demonstrated in mice. Coming to a similar set of conclusions in human NPCs in vitro seems fairly incremental.

RE: Thanks for this suggestion. Yes, we agree that these reports had provided findings that JMJD3 and UTX play important roles in neural development based on mouse model. However, we think our analysis with human model did provide novel mechanisms on KDM6 function during neurogenesis. Based on the human ES neural differentiation model that involves serial fate transition from pluripotent stem cells(PSCs) to neural progenitor cells(NPCs) and then neurons and glia cells, we revealed a differential requirement on KDM6s at different stage of neuro/glia genesis, in NPCs and neuron fate transition. Mechanistically, the NPC specific BAF units (npBAF) that promote NPC specification do not require KDM6 mediated H3K27me3 demethylation, while neuron specific BAFs (nBAF) definitely need KDM6s for their proper expression (Figure 7d and Figure 7b). In addition, based on the new data in revised version, KDM6s are important to maintain protein stability of BAF core units (Figure 7a) at later stage of neuron differentiation. Together with other reports (as responded “above”) , these data demonstrated an inter-dependent interaction between KDM6s and BAFs in neurogenesis. Lastly, we also revealed that nBAFs rescued neuron/glia defect in dKO cells independent of H3K27me3 through re-establishing the

permissive chromatin state in NPCs (Figure 7e-g). As agreed by all three reviewers, how chromatin-modifying factors control fate transition in neuron/glia genesis is a significant interesting question. Here, we provided a detailed analysis on how KDM6s and BAFs work together to control the fate transition from PSCs to NPCs and then subtype of neural cells. To our knowledge and based on the literatures listed by the reviewer here, these analyses had not been done in those mouse model, thus hope to be appreciated by the reviewer.

The genome-wide analyses (H3K27me3 ChIP-seq, RNA-seq, and ATAC-seq) of JMJD3 and UTX knockouts should be complemented by JMJD3 and UTX ChIP-seq. Without knowledge of where JMJD3 and UTX are localized on the genome, it is difficult to interpret the chromatin and transcriptomic data presented in this manuscript. That is, simply observing increased levels of H3K27me3 and decreased accessibility at a particular locus does not demonstrate that JMJD3 and/or UTX normally functions at that locus. For instance, it is possible that EZH2 (the H3K27me3 methyltransferase) activity at those genes increases with KDM6-deficiency. Of note, JMJD3 ChIP-seq has been achieved in studies of mouse NSCs (e.g., Fueyo et al., Nuc Acid Res, 2018; Estaras et al., MCB 2013), which has helped our understanding of JMJD3 mechanism(s) in transcriptional activation. For instance, in addition to demethylation of H3K27me3 at promoters, KDM6s have important function at enhancers as well as in transcriptional elongation. These additional mechanisms should be considered.

RE: Thanks for this suggestion. We actually made a lot of efforts to address this issue since even though we tried many times, it's still hard to find a good antibody for JMJD3 or UTX ChIP-seq. We agree this is an important issue and thus knocked-in a triple-FLAG tag to C terminal of UTX through gene targeting. After careful characterizations, we performed ChIP-seq using FLAG antibody in NPCs (New Figure 6d). Consistent to other data in the manuscript, genes bound by UTX are significantly enriched in the function of neurogenesis, cell cycle transition, etc., important to neurogenesis. Also, UTX bound genes showed much more reduced H3K27me3 enrichment and more accessible chromatin (Figure 6e-f). These data further support the critical role of KDM6s in neurogenesis and hope these data will satisfy.

Other comments:

1. Does knockdown of BAF53a in dKO cells rescue neurogenesis?

RE: Thanks for this suggestion. We performed experiment on BAF53A knock-down and found it indeed can rescued differentiation defect in KDM6 deficient cells (Supplementary Figure 5d), indicating a direct antagonism between npBAF and nBAF in neurogenesis.

2. When it is overexpressed, does BAF53b localize to expected targets? Or, does BAF53b simply put the entire cell into a generally more transcriptionally permissive state?

RE: Thanks for this suggestion. Yes, this is a good question. We performed additional

ATAC-seq analysis in WT-NPCs, dKO-NPCs, and BAF53B-rescued dKO-NPCs. We showed that nBAFs rescued neuron/glia defect in dKO cells independent of H3K27me3 through re-establishing the permissive chromatin state for neurogenesis in NPCs (Figure 7e-g).

Reviewers' comments:

Reviewer #1 (Remarks to the Author):

The authors have addressed my concerns. I have no further comments.

Reviewer #2 (Remarks to the Author):

The revised manuscript has been significantly improved with clarification to the text. I have no further concerns.

Reviewer #3 (Remarks to the Author):

The authors have performed new experiments and analyses to address reviewer comments. However, despite their notable efforts, some of my major concerns remain. In this paper, the authors have studied JMJD3-KO, UTX-KO, and JMJD3-KO/UTX-KO double knockout (dKO) human ESCs. UTX and JMJD3 are H3K27me3-demethylases. Most of the experiments were performed with the dKO cells. In the dKO cells, many genes accumulated H3K27me3, but this finding can be difficult to interpret without knowledge of where JMJD3 and UTX are located across the genome. For instance, are certain genes targeted by only JMJD3 or UTX? Are some genes targeted by both? Or, perhaps some genes do not exhibit enrichment of either JMJD3 or UTX, making it possible that these are more directly regulated by EZH2 (H3K27me3 methyltransferase). The authors were able to assess UTX localization with ChIP-seq in new experiments, but they do not have JMJD3 localization data. Thus, it is still very difficult to interpret data from the dKO cells.

The part of the story that revolves around BAF subunits is also still unclear. In dKO cells, the mRNA levels of BRG1 and BRM are not changed, but protein levels are decreased. The authors suggest that JMJD3 and/or UTX "...promote protein stability of core BAF subunits." Proteasome inhibitor treatment resulted in more normal levels of BRG1 and BRM proteins, but how JMJD3/UTX might contribute to inhibition of BRG1/BRM degradation is not shown. It does not seem that JMJD3/UTX regulate the chromatin state of BAF genes. These results led the authors to this vague conclusion: "These data demonstrate that KDM6s are differentially required in regulating BAF subunit genes." However, none of the regulation of these BAF subunits appears direct.

Reviewer #1 (Remarks to the Author):

The authors have addressed my concerns. I have no further comments.

RE: We thank the reviewer for the positive comments.

Reviewer #2 (Remarks to the Author):

The revised manuscript has been significantly improved with clarification to the text. I have no further concerns.

RE: We thank the reviewer for the positive comments.

Reviewer #3 (Remarks to the Author):

The authors have performed new experiments and analyses to address reviewer comments. However, despite their notable efforts, some of my major concerns remain. In this paper, the authors have studied JMJD3-KO, UTX-KO, and JMJD3-KO/UTX-KO double knockout (dKO) human ESCs. UTX and JMJD3 are H3K27me3-demethylases. Most of the experiments were performed with the dKO cells. In the dKO cells, many genes accumulated H3K27me3, but this finding can be difficult to interpret without knowledge of where JMJD3 and UTX are located across the genome. For instance, are certain genes targeted by only JMJD3 or UTX? Are some genes targeted by both? Or, perhaps some genes do not exhibit enrichment of either JMJD3 or UTX, making it possible that these are more directly regulated by EZH2 (H3K27me3 methyltransferase). The authors were able to assess UTX localization with ChIP-seq in new experiments, but they do not have JMJD3 localization data. Thus, it is still very difficult to interpret data from the dKO cells.

RE: Thanks for these comments. To address them, we performed a lot more experiments to generate another hESCs with a triple-FLAG knock-in in JMJD3 and performed ChIP-seq for genome mapping on JMJD3 location in human NPCs. Firstly, JMJD3 localization could be detected on almost all gene promoters bound by UTX in hNPCs. As expected, these co-bound genes by JMJD3/UTX are extremely related to neurogenesis and neuron differentiation. However, we did detect a set of genes showing high JMJD3 bindings while very low or no UTX bindings. These JMJD3-only binding genes are related to autophagy function, indicating the additional role of JMJD3 in hNPCs compared with UTX. Anyways, our new data clearly demonstrates that the major functions of JMJD3 and UTX in hNPCs are to regulate neuron specification, consistent to the differentiation defect observed in dKO cells.

The part of the story that revolves around BAF subunits is also still unclear. In dKO cells, the mRNA levels of BRG1 and BRM are not changed, but protein levels are decreased. The authors suggest that JMJD3 and/or UTX "...promote protein stability of core BAF subunits." Proteasome inhibitor treatment resulted in more normal levels of BRG1 and BRM proteins, but how JMJD3/UTX might contribute to inhibition of BRG1/BRM degradation is not shown. It does not seem that JMJD3/UTX regulate the chromatin state of BAF genes. These results led the authors to this vague conclusion: "These data demonstrate that KDM6s are differentially required in regulating BAF subunit genes." However, none of the regulation of these BAF subunits appears direct.

RE: Thanks for this suggestion. The degradation of BRG1/BRM was rescued by proteasome inhibitor, MG132(Supplemental Figure 7a), indicating that the degradation mechanism was through proteasome. Based on previous reports (Reference 52) and our results, UTX/JMJD3 might form a big complex with BAF and help to promote the stability of the whole complex. It's an interesting phenomena and worth to peruse the detail further, but beyond the scope current manuscript. Accordingly, we revised the text on some conclusions as suggested.

** See Nature Research's author and referees' website at www.nature.com/authors for information about policies, services and author benefits

REVIEWERS' COMMENTS:

Reviewer #3 (Remarks to the Author):

The authors have been very responsive to my comment about the need for JMJD3 localization data. The new FLAG-JMJD3 ChIP-seq data is a nice addition and helps understand the phenotype of the KDM6-dKO cells. A minor comment is that some of the text in Figures (particularly the GO term listings) are in a font that is a bit too small to read. Also, given that this paper concerns neurogenesis and H3K27me3 de-methylation, it seems that in the Introduction and/or Discussion, the authors should discuss their results in the context of related work done in mice (the overall model that H3K27me3 de-methylation by UTX (Lei and Jiao, Stem Cell Rep 2018) and JMJD3 (Park, Cell Rep 2014) at specific genes/enhancers is important to neurogenesis has been previously shown in mice), so that the results can be interpreted into the broader in vivo context.

REVIEWERS' COMMENTS:

Reviewer #3 (Remarks to the Author):

The authors have been very responsive to my comment about the need for JMJD3 localization data. The new FLAG-JMJD3 ChIP-seq data is a nice addition and helps understand the phenotype of the KDM6-dKO cells. A minor comment is that some of the text in Figures (particularly the GO term listings) are in a font that is a bit too small to read. Also, given that this paper concerns neurogenesis and H3K27me3 de-methylation, it seems that in the Introduction and/or Discussion, the authors should discuss their results in the context of related work done in mice (the overall model that H3K27me3 de-methylation by UTX (Lei and Jiao, Stem Cell Rep 2018) and JMJD3 (Park, Cell Rep 2014) at specific genes/enhancers is important to neurogenesis has been previously shown in mice), so that the results can be interpreted into the broader in vivo context.

Response: Thanks for your suggestion. We have updated the big font for words in some of the text in Figures including the GO term listings. Next, we cited and discussed previous related work done in mice in the Introduction and Discussion sections. Please see the revised manuscript.